# Is Conditional Generative Modeling all you need for Decision-Making?

**Anurag Ajay**[*†§¶], **Yilun Du**[*§¶], **Abhi Gupta**[*‡§¶], **Joshua Tenenbaum**[¶], **Tommi Jaakkola**[‡§¶], **Pulkit Agrawal**[†§¶]

Improbable AI Lab[†]
Operations Research Center[‡]
Computer Science and Artificial Intelligence Lab[§]
Massachusetts Institute of Technology[¶]

## ABSTRACT

Recent improvements in conditional generative modeling have made it possible to generate high-quality images from language descriptions alone. We investigate whether these methods can directly address the problem of sequential decision-making. We view decision-making not through the lens of reinforcement learning (RL), but rather through conditional generative modeling. To our surprise, we find that our formulation leads to policies that can outperform existing offline RL approaches across standard benchmarks. By modeling a policy as a return-conditional diffusion model, we illustrate how we may circumvent the need for dynamic programming and subsequently eliminate many of the complexities that come with traditional offline RL. We further demonstrate the advantages of modeling policies as conditional diffusion models by considering two other conditioning variables: constraints and skills. Conditioning on a single constraint or skill during training leads to behaviors at test-time that can satisfy several constraints together or demonstrate a composition of skills. Our results illustrate that conditional generative modeling is a powerful tool for decision-making.

## 1 INTRODUCTION

Over the last few years, conditional generative modeling has yielded impressive results in a range of domains, including high-resolution image generation from text descriptions (DALL-E, ImageGen) (Ramesh et al., 2022; Saharia et al., 2022), language generation (GPT) (Brown et al., 2020), and step-by-step solutions to math problems (Minerva) (Lewkowycz et al., 2022). The success of generative models in countless domains motivates us to apply them to decision-making. Conveniently, there exists a wide body of research on recovering high-performing policies from data logged by already operational systems (Kostrikov et al., 2022; Kumar et al., 2020; Walke et al., 2022). This is particularly useful in real-world settings where interacting with the environment is not always possible, and exploratory decisions can have fatal consequences (Dulac-Arnold et al., 2021). With access to such offline datasets, the problem of decision-making reduces to learning a probabilistic model of trajectories, a setting where generative models have already found success.

In offline decision-making, we aim to recover optimal reward-maximizing trajectories by stitching together sub-optimal reward-labeled trajectories in the training dataset. Prior works (Kumar et al., 2020; Kostrikov et al., 2022; Wu et al., 2019; Kostrikov et al., 2021; Dadashi et al., 2021; Ajay et al., 2020; Ghosh et al., 2022) have tackled this problem with reinforcement learning (RL) that uses dynamic programming for trajectory stitching. To enable dynamic programming, these works learn a *value* function that estimates the discounted sum of rewards from a given state. However, value function estimation is prone to instabilities due to function approximation, off-policy learning, and bootstrapping together, together known as the *deadly triad* (Sutton & Barto, 2018). Furthermore, to stabilize value estimation in offline regime, these works rely on heuristics to keep the policy within the dataset distribution. These challenges make it difficult to scale existing offline RL algorithms.

---

* denotes equal contribution. Correspondence to `aajay@mit.edu`, `yilundu@mit.edu`, `abhig@mit.edu`

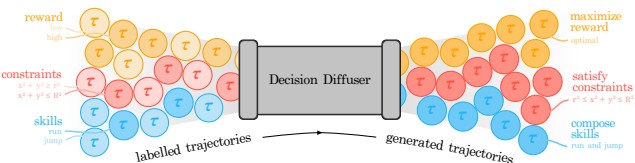

Figure 1: **Decision Making using Conditional Generative Modeling.** Framing decision making as a conditional generative modeling problem allows us to maximize rewards, satisfy constraints and compose skills.

In this paper, we ask if we can perform dynamic programming to stitch together sub-optimal trajectories to obtain an optimal trajectory without relying on value estimation. Since conditional diffusion generative models can generate novel data points by composing training data (Saharia et al., 2022), we leverage it for trajectory stitching in offline decision-making. Given a dataset of reward-labeled trajectories, we adapt diffusion models (Sohl-Dickstein et al., 2015) to learn a return-conditional trajectory model. During inference, we use *classifier-free guidance* with *low-temperature sampling*, which we hypothesize to implicitly perform dynamics programming, to capture the best behaviors in the dataset and glean return maximizing trajectories (see Appendix A). Our straightforward conditional generative modeling formulation outperforms existing approaches on standard D4RL tasks (Fu et al., 2020).

Viewing offline decision-making through the lens of conditional generative modeling allows going beyond conditioning on returns (Figure 1). Consider an example (detailed in Appendix A) where a robot with linear dynamics navigates an environment containing two concentric circles (Figure 2). We are given a dataset of state-action trajectories of the robot, each satisfying one of two constraints: (i) the final position of the robot is within the larger circle, and (ii) the final position of the robot is outside the smaller circle. With conditional diffusion modeling, we can use the datasets to learn a constraint-conditioned model that can generate trajectories satisfying any set of constraints. During inference, the learned trajectory model can merge constraints from the dataset and generate trajectories that satisfy the combined constraint. Figure 2 shows that the constraint-conditioned model can generate trajectories such that the final position of the robot lies between the concentric circles.

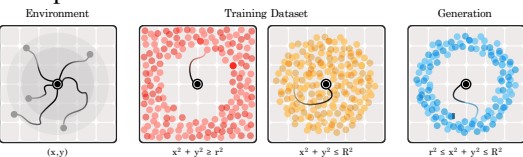

Figure 2: **Illustrative example.** We visualize the 2d robot navigation environment and the constraints satisfied by the trajectories in the dataset derived from the environment. We show the ability of the conditional diffusion model to generate trajectories that satisfy the combined constraints.

Here, we demonstrate the benefits of modeling policies as conditional generative models. First, conditioning on constraints allows policies to not only generate behaviors satisfying individual constraints but also generate novel behaviors by flexibly combining constraints at test time. Further, conditioning on skills allows policies to not only imitate individual skills but also generate novel behaviors by composing those skills. We instantiate this idea with a state-sequence based diffusion probabilistic model (Ho et al., 2020) called *Decision Diffuser*, visualized in Figure 1. In summary, our contributions include **(i)** illustrating conditional generative modeling as an effective tool in offline decision making, **(ii)** using classifier-free guidance with low-temperature sampling, instead of dynamic programming, to get return-maximizing trajectories and, **(iii)** leveraging the framework of conditional generative modeling to combine constraints and compose skills during inference flexibly.

## 2 BACKGROUND

### 2.1 REINFORCEMENT LEARNING

We formulate the sequential decision-making problem as a discounted Markov Decision Process (MDP) defined by the tuple $\langle \rho_0, \mathcal{S}, \mathcal{A}, \mathcal{T}, \mathcal{R}, \gamma \rangle$, where $\rho_0$ is the initial state distribution, $\mathcal{S}$ and $\mathcal{A}$ are state and action spaces, $\mathcal{T} : \mathcal{S} \times \mathcal{A} \to \mathcal{S}$ is the transition function, $\mathcal{R} : \mathcal{S} \times \mathcal{A} \times \mathcal{S} \to \mathbb{R}$ gives the reward at any transition and $\gamma \in [0, 1)$ is a discount factor. The agent acts with a stochastic policy $\pi : \mathcal{S} \to \Delta_{\mathcal{A}}$, generating a sequence of state-action-reward transitions or trajectory $\tau := (s_k, a_k, r_k)_{k \geq 0}$ with probability $p_\pi(\tau)$ and return $R(\tau) := \sum_{k \geq 0} \gamma^k r_k$. The standard objective in RL is to find a return-maximizing policy $\pi^* = \arg\max_\pi \mathbb{E}_{\tau \sim p_\pi}[R(\tau)]$.

**Temporal Difference Learning**    TD methods (Fujimoto et al., 2018; Lillicrap et al., 2015) estimate $Q^*(s, a) \coloneqq \mathbb{E}_{\tau \sim p_{\pi^*}}[R(\tau)|s_0 = s, a_0 = a]$, the return achieved under the optimal policy $\pi^*$ when starting in state $s$ and taking action $a$, with a parameterized $Q$-function. This requires minimizing the following TD loss:

$$\mathcal{L}_{\text{TD}}(\theta) \coloneqq \mathbb{E}_{(s,a,r,s') \in \mathcal{D}}[(r + \gamma \max_{a' \in \mathcal{A}} Q_\theta(s', a') - Q_\theta(s, a))^2] \tag{1}$$

Continuous action spaces further require learning a parametric policy $\pi_\phi(a|s)$ that plays the role of the maximizing action in equation 1. This results in a policy objective that must be maximized:

$$\mathcal{J}(\phi) \coloneqq \mathbb{E}_{s \in \mathcal{D}, a \sim \pi_\phi(\cdot|s)}[Q(s, a)] \tag{2}$$

Here, the dataset of transitions $\mathcal{D}$ evolves as the agent interacts with the environment and both $Q_\theta$ and $\pi_\phi$ are trained together. These methods make use of function approximation, off-policy learning, and bootstrapping, leading to several instabilities in practice (Sutton, 1988; Van Hasselt et al., 2018).

**Offline RL**    requires finding a return-maximizing policy from a fixed dataset of transitions collected by an unknown behavior policy $\mu$ (Levine et al., 2020). Using TD-learning naively causes the state visitation distribution $d^{\pi_\phi}(s)$ to move away from the distribution of the dataset $d^\mu(s)$. In turn, the policy $\pi_\phi$ begins to take actions that are substantially different from those already seen in the data. Offline RL algorithms resolve this distribution-shift by imposing a constraint of the form $D(d^{\pi_\phi}||d^\mu)$, where $D$ is some divergence metric, directly in the TD-learning procedure. The constrained optimization problem now demands additional implementation heuristics to achieve any reasonable performance (Kumar et al., 2021). The Decision Diffuser, in comparison, doesn't have any of these disadvantages. It does not require estimating any kind of $Q$-function, thereby sidestepping TD methods altogether. It also does not face the risk of distribution-shift as generative models are trained with maximum-likelihood estimation.

## 2.2    DIFFUSION PROBABILISTIC MODELS

Diffusion models (Sohl-Dickstein et al., 2015; Ho et al., 2020) are a specific type of generative model that learn the data distribution $q(\boldsymbol{x})$ from a dataset $\mathcal{D} \coloneqq \{\boldsymbol{x}^i\}_{0 \leq i < M}$. They have been used most notably for synthesizing high-quality images from text descriptions (Saharia et al., 2022; Nichol et al., 2021). Here, the data-generating procedure is modelled with a predefined forward noising process $q(\boldsymbol{x}_{k+1}|\boldsymbol{x}_k) \coloneqq \mathcal{N}(\boldsymbol{x}_{k+1}; \sqrt{\alpha_k}\boldsymbol{x}_k, (1 - \alpha_k)\boldsymbol{I})$ and a trainable reverse process $p_\theta(\boldsymbol{x}_{k-1}|\boldsymbol{x}_k) \coloneqq \mathcal{N}(\boldsymbol{x}_{k-1}|\mu_\theta(\boldsymbol{x}_k, k), \Sigma_k)$, where $\mathcal{N}(\mu, \Sigma)$ denotes a Gaussian distribution with mean $\mu$ and variance $\Sigma$, $\alpha_k \in \mathbb{R}$ determines the variance schedule, $\boldsymbol{x}_0 \coloneqq \boldsymbol{x}$ is a sample, $\boldsymbol{x}_1, \boldsymbol{x}_2, ..., \boldsymbol{x}_{K-1}$ are the latents, and $\boldsymbol{x}_K \sim \mathcal{N}(\boldsymbol{0}, \boldsymbol{I})$ for carefully chosen $\alpha_k$ and long enough $K$. Starting with Gaussian noise, samples are then iteratively generated through a series of "denoising" steps.

Although a tractable variational lower-bound on $\log p_\theta$ can be optimized to train diffusion models, Ho et al. (2020) propose a simplified surrogate loss:

$$\mathcal{L}_{\text{denoise}}(\theta) \coloneqq \mathbb{E}_{k \sim [1,K], \boldsymbol{x}_0 \sim q, \epsilon \sim \mathcal{N}(\boldsymbol{0}, \boldsymbol{I})}[||\epsilon - \epsilon_\theta(\boldsymbol{x}_k, k)||^2] \tag{3}$$

The predicted noise $\epsilon_\theta(\boldsymbol{x}_k, k)$, parameterized with a deep neural network, estimates the noise $\epsilon \sim \mathcal{N}(0, I)$ added to the dataset sample $\boldsymbol{x}_0$ to produce noisy $\boldsymbol{x}_k$. This is equivalent to predicting the mean of $p_\theta(\boldsymbol{x}_{k-1}|\boldsymbol{x}_k)$ since $\mu_\theta(\boldsymbol{x}_k, k)$ can be calculated as a function of $\epsilon_\theta(\boldsymbol{x}_k, k)$ (Ho et al., 2020).

**Guided Diffusion**    Modelling the conditional data distribution $q(\boldsymbol{x}|\boldsymbol{y})$ makes it possible to generate samples with attributes of the label $\boldsymbol{y}$. The equivalence between diffusion models and score-matching (Song et al., 2021), which shows $\epsilon_\theta(\boldsymbol{x}_k, k) \propto \nabla_{\boldsymbol{x}_k} \log p(\boldsymbol{x}_k)$, leads to two kinds of methods for conditioning: classifier-guided (Nichol & Dhariwal, 2021) and classifier-free (Ho & Salimans, 2022). The former requires training an additional classifier $p_\phi(\boldsymbol{y}|\boldsymbol{x}_k)$ on noisy data so that samples may be generated at test-time with the perturbed noise $\epsilon_\theta(\boldsymbol{x}_k, k) - \omega\sqrt{1 - \bar{\alpha}_k}\nabla_{\boldsymbol{x}_k} \log p(\boldsymbol{y}|\boldsymbol{x}_k)$, where $\omega$ is referred to as the guidance scale. The latter does not separately train a classifier but modifies the original training setup to learn both a conditional $\epsilon_\theta(\boldsymbol{x}_k, \boldsymbol{y}, k)$ and an unconditional $\epsilon_\theta(\boldsymbol{x}_k, k)$ model for the noise. The unconditional noise is represented, in practice, as the conditional noise $\epsilon_\theta(\boldsymbol{x}_k, \varnothing, k)$ where a dummy value $\varnothing$ takes the place of $\boldsymbol{y}$. The perturbed noise $\epsilon_\theta(\boldsymbol{x}_k, k) + \omega(\epsilon_\theta(\boldsymbol{x}_k, \boldsymbol{y}, k) - \epsilon_\theta(\boldsymbol{x}_k, k))$ is used to later generate samples.

## 3    GENERATIVE MODELING WITH THE DECISION DIFFUSER

It is useful to solve RL from offline data, both without relying on TD-learning and without risking distribution-shift. To this end, we formulate sequential decision-making as the standard problem of

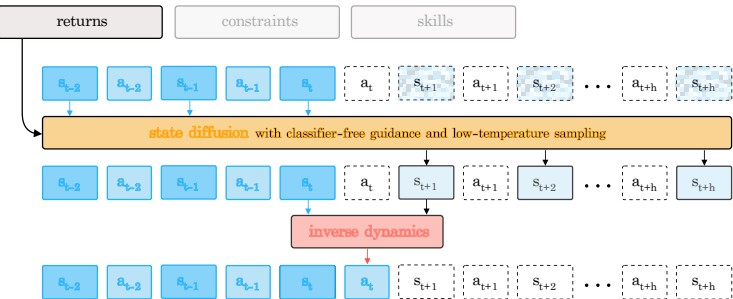

Figure 3: **Planning with Decision Diffuser.** Given the current state $s_t$ and conditioning, Decision Diffuser uses classifier-free guidance with low-temperature sampling to generate a sequence of future states. It then uses inverse dynamics to extract and execute the action $a_t$ that leads to the immediate future state $s_{t+1}$.

conditional generative modeling:

$$\max_\theta \mathbb{E}_{\tau \sim \mathcal{D}}[\log p_\theta(\boldsymbol{x}_0(\tau)|\boldsymbol{y}(\tau))] \tag{4}$$

Our goal is to estimate the conditional data distribution with $p_\theta$ so we can later generate portions of a trajectory $\boldsymbol{x}_0(\tau)$ from information $\boldsymbol{y}(\tau)$ about it. Examples of $\boldsymbol{y}$ could include the return under the trajectory, the constraints satisfied by the trajectory, or the skill demonstrated in the trajectory. We construct our generative model according to the conditional diffusion process:

$$q(\boldsymbol{x}_{k+1}(\tau)|\boldsymbol{x}_k(\tau)), \quad p_\theta(\boldsymbol{x}_{k-1}(\tau)|\boldsymbol{x}_k(\tau), \boldsymbol{y}(\tau)) \tag{5}$$

As usual, $q$ represents the forward noising process while $p_\theta$ the reverse denoising process. In the following, we discuss how we may use diffusion for decision making. First, we discuss the modeling choices for diffusion in Section 3.1. Next, we discuss how we may utilize classifier-free guidance to capture the best aspects of trajectories in Section 3.2. We then discuss the different behaviors that may be implemented with conditional diffusion models in Section 3.3. Finally, we discuss practical training details of our approach in Section 3.4.

## 3.1 DIFFUSING OVER STATES

In images, the diffusion process is applied across all pixel values in an image. Naïvely, it would therefore be natural to apply a similar process to model the state and actions of a trajectory. However, in the reinforcement learning setting, directly modeling actions using a diffusion process has several practical issues. First, while states are typically continuous in nature in RL, actions are more varied, and are often discrete in nature. Furthermore, sequences over actions, which are often represented as joint torques, tend to be more high-frequency and less smooth, making them much harder to predict and model (Tedrake, 2022). Due to these practical issues, we choose to diffuse only over states, as defined below:

$$\boldsymbol{x}_k(\tau) := (s_t, s_{t+1}, ..., s_{t+H-1})_k \tag{6}$$

Here, $k$ denotes the timestep in the forward process and $t$ denotes the time at which a state was visited in trajectory $\tau$. Moving forward, we will view $\boldsymbol{x}_k(\tau)$ as a noisy sequence of states from a trajectory of length $H$. We represent $\boldsymbol{x}_k(\tau)$ as a two-dimensional array with one column for each timestep of the sequence.

**Acting with Inverse-Dynamics.** Sampling states from a diffusion model is not enough for defining a controller. A policy can, however, be inferred from estimating the action $a_t$ that led the state $s_t$ to $s_{t+1}$ for any timestep $t$ in $\boldsymbol{x}_0(\tau)$. Given two consecutive states, we generate an action according to the inverse dynamics model (Agrawal et al., 2016; Pathak et al., 2018):

$$a_t := f_\phi(s_t, s_{t+1}) \tag{7}$$

Note that the same offline data used to train the reverse process $p_\theta$ can also be used to learn $f_\phi$. We illustrate in Table 2 how the design choice of directly diffusing state distributions, with an inverse dynamics model to predict action, significantly improves performance over diffusing across both states and actions jointly. Furthermore, we empirically compare and analyze when to use inverse dynamics and when to diffuse over actions in Appendix F.

## 3.2 Planning with Classifier-Free Guidance

Given a diffusion model representing the different trajectories in a dataset, we next discuss how we may utilize the diffusion model for planning. To use the model for planning, it is necessary to additionally condition the diffusion process on characteristics $\boldsymbol{y}(\tau)$. One approach could be to train a classifier $p_\phi(\boldsymbol{y}(\tau)|\boldsymbol{x}_k(\tau))$ to predict $\boldsymbol{y}(\tau)$ from noisy trajectories $\boldsymbol{x}_k(\tau)$. In the case that $\boldsymbol{y}(\tau)$ represents the return under a trajectory, this would require estimating a $Q$-function, which requires a separate, complex dynamic programming procedure.

One approach to avoid dynamic programming is to directly train a conditional diffusion model conditioned on the returns $\boldsymbol{y}(\tau)$ in the offline dataset. However, as our dataset consists of a set of sub-optimal trajectories, the conditional diffusion model will be polluted by such sub-optimal behaviors. To circumvent this issue, we utilize classifier-free guidance (Ho & Salimans, 2022) with low-temperature sampling, to extract high-likelihood trajectories in the dataset. We find that such trajectories correspond to the best set of behaviors in the dataset. For a detailed discussion comparing Q-function guidance and classifier-free guidance, please refer to Appendix K. Formally, to implement classifier free guidance, a $\boldsymbol{x}_0(\tau)$ is sampled by starting with Gaussian noise $\boldsymbol{x}_K(\tau)$ and refining $\boldsymbol{x}_k(\tau)$ into $\boldsymbol{x}_{k-1}(\tau)$ at each intermediate timestep with the perturbed noise:

$$\hat{\epsilon} := \epsilon_\theta(\boldsymbol{x}_k(\tau), \varnothing, k) + \omega(\epsilon_\theta(\boldsymbol{x}_k(\tau), \boldsymbol{y}(\tau), k) - \epsilon_\theta(\boldsymbol{x}_k(\tau), \varnothing, k)), \tag{8}$$

where the scalar $\omega$ applied to $(\epsilon_\theta(\boldsymbol{x}_k(\tau), \boldsymbol{y}(\tau), k) - \epsilon_\theta(\boldsymbol{x}_k(\tau), \varnothing, k))$ seeks to augment and extract the best portions of trajectories in the dataset that exhibit $\boldsymbol{y}(\tau)$. With these ingredients, sampling from the Decision Diffuser becomes similar to planning in RL. First, we observe a state in the environment. Next, we sample states later into the horizon with our diffusion process conditioned on $\boldsymbol{y}$ and history of last $C$ states observed. Finally, we identify the action that should be taken to reach the most immediate predicted state with our inverse dynamics model. This procedure repeats in a standard receding-horizon control loop described in Algorithm 1 and visualized in Figure 3.

## 3.3 Conditioning beyond Returns

So far we have not explicitly defined the conditioning variable $\boldsymbol{y}(\tau)$. Though we have mentioned that it can be the return under a trajectory, we may also consider guiding our diffusion process towards sequences of states that satisfy relevant constraints or demonstrate specific behavior.

**Maximizing Returns** To generate trajectories that maximize return, we condition the noise model on the return of a trajectory so $\epsilon_\theta(\boldsymbol{x}_k(\tau), \boldsymbol{y}(\tau), k) := \epsilon_\theta(\boldsymbol{x}_k(\tau), R(\tau), k)$. These returns are normalized to keep $R(\tau) \in [0, 1]$. Sampling a high return trajectory amounts to conditioning on $R(\tau) = 1$. Note that we do not make use of any $Q$-values, which would then require dynamic programming.

**Satisfying Constraints** Trajectories may satisfy a variety of constraints, each represented by the set $\mathcal{C}_i$, such as reaching a specific goal, visiting states in a particular order, or avoiding parts of the state space. To generate trajectories satisfying a given constraint $\mathcal{C}_i$, we condition the noise model on a one-hot encoding so that $\epsilon_\theta(\boldsymbol{x}_k(\tau), \boldsymbol{y}(\tau), k) := \epsilon_\theta(\boldsymbol{x}_k(\tau), \mathbb{1}(\tau \in \mathcal{C}_i), k)$. Although we train with an offline dataset in which trajectories satisfy only one of the available constraints, at inference we can satisfy several constraints together.

**Composing Skills** A skill $i$ can be specified from a set of demonstrations $\mathcal{B}_i$. To generate trajectories that demonstrate a given skill, we condition the noise model on a one-hot encoding so that $\epsilon_\theta(\boldsymbol{x}_k(\tau), \boldsymbol{y}(\tau), k) := \epsilon_\theta(\boldsymbol{x}_k(\tau), \mathbb{1}(\tau \in \mathcal{B}_i), k)$. Although we train with individual skills, we may further compose these skills together during inference.

Assuming we have learned the data distributions $q(\boldsymbol{x}_0(\tau)|\boldsymbol{y}^1(\tau)), \ldots, q(\boldsymbol{x}_0(\tau)|\boldsymbol{y}^n(\tau))$ for $n$ different conditioning variables, we can sample from the composed data distribution $q(\boldsymbol{x}_0(\tau)|\boldsymbol{y}^1(\tau), \ldots, \boldsymbol{y}^n(\tau))$ using the perturbed noise (Liu et al., 2022):

$$\hat{\epsilon} := \epsilon_\theta(\boldsymbol{x}_k(\tau), \varnothing, k) + \omega \sum_{i=1}^{n} (\epsilon_\theta(\boldsymbol{x}_k(\tau), \boldsymbol{y}^i(\tau), k) - \epsilon_\theta(\boldsymbol{x}_k(\tau), \varnothing, k)) \tag{9}$$

This property assumes that $\{\boldsymbol{y}^i(\tau)\}_{i=1}^{n}$ are conditionally independent given the state trajectory $\boldsymbol{x}_0(\tau)$. However, we empirically observe that this assumption doesn't have to be strictly satisfied as long as the composition of conditioning variables is feasible. For more detailed discussion, please refer to Appendix D. We use this property to compose more than one constraint or skill together at test-time. We also show how Decision Diffuser can avoid particular constraint or skill (NOT) in Appendix J.

**Algorithm 1** Conditional Planning with the Decision Diffuser

1: **Input:** Noise model $\epsilon_\theta$, inverse dynamics $f_\phi$, guidance scale $\omega$, history length $C$, condition $\boldsymbol{y}$
2: Initialize $h \leftarrow \texttt{Queue}(\texttt{length} = C), t \leftarrow 0$        // Maintain a history of length C
3: **while** not done **do**
4:  Observe state $s$; $h.\texttt{insert}(s)$; Initialize $\boldsymbol{x}_K(\tau) \sim \mathcal{N}(0, \alpha I)$
5:  **for** $k = K \dots 1$ **do**
6:   $\boldsymbol{x}_k(\tau)[: \texttt{length}(h)] \leftarrow h$      // Constrain plan to be consistent with history
7:   $\hat{\epsilon} \leftarrow \epsilon_\theta(\boldsymbol{x}_k(\tau), k) + \omega(\epsilon_\theta(\boldsymbol{x}_k(\tau), \boldsymbol{y}, k) - \epsilon_\theta(\boldsymbol{x}_k(\tau), k))$   // Classifier-free guidance
8:   $(\mu_{k-1}, \Sigma_{k-1}) \leftarrow \texttt{Denoise}(\boldsymbol{x}_k(\tau), \hat{\epsilon})$
9:   $\boldsymbol{x}_{k-1} \sim \mathcal{N}(\mu_{k-1}, \alpha\Sigma_{k-1})$
10:  **end for**
11: Extract $(s_t, s_{t+1})$ from $x_0(\tau)$
12: Execute $a_t = f_\phi(s_t, s_{t+1})$; $t \leftarrow t + 1$
13: **end while**

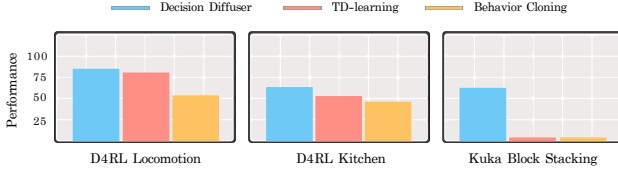

Figure 4: **Results Overview.** Decision Diffuser performs better than both TD learning (`CQL`) and Behavorial Cloning (`BC`) across D4RL locomotion tasks, D4RL Kitchen tasks and Kuka Block Stacking tasks (single constraint) using only a conditional generative modeling objective. For performance metric, we use normalized average returns (Fu et al., 2020) for D4RL tasks (Locomotion and Kitchen) and success rate for Block Stacking.

## 3.4 Training the Decision Diffuser

The Decision Diffuser, our conditional generative model for decision-making, is trained in a supervised manner. Given a dataset $\mathcal{D}$ of trajectories, each labeled with the return it achieves, the constraint that it satisfies, or the skill that it demonstrates, we simultaneously train the reverse diffusion process $p_\theta$, parameterized through the noise model $\epsilon_\theta$, and the inverse dynamics model $f_\phi$ with the following loss:

$$\mathcal{L}(\theta, \phi) := \mathbb{E}_{k, \tau \in \mathcal{D}, \beta \sim \text{Bern}(p)}[||\epsilon - \epsilon_\theta(\boldsymbol{x}_k(\tau), (1-\beta)\boldsymbol{y}(\tau) + \beta\varnothing, k)||^2] + \mathbb{E}_{(s, a, s') \in \mathcal{D}}[||a - f_\phi(s, s')||^2]$$

For each trajectory $\tau$, we first sample noise $\epsilon \sim \mathcal{N}(\boldsymbol{0}, \boldsymbol{I})$ and a timestep $k \sim \mathcal{U}\{1, \dots, K\}$. Then, we construct a noisy array of states $\boldsymbol{x}_k(\tau)$ and finally predict the noise as $\hat{\epsilon}_\theta := \epsilon_\theta(\boldsymbol{x}_k(\tau), \boldsymbol{y}(\tau), k)$. Note that with probability $p$ we ignore the conditioning information and the inverse dynamics is trained with individual transitions rather than trajectories.

**Architecture** We parameterize $\epsilon_\theta$ with a temporal U-Net architecture, a neural network consisting of repeated convolutional residual blocks (Janner et al., 2022). This effectively treats a sequence of states $\boldsymbol{x}_k(\tau)$ as an image where the height represents the dimension of a single state and the width denotes the length of the trajectory. We encode the conditioning information $\boldsymbol{y}(\tau)$ as either a scalar or a one-hot vector and project it into a latent variable $z \in \mathbb{R}^h$ with a multi-layer perceptron (MLP). When $\boldsymbol{y}(\tau) = \varnothing$, we zero out the entries of $z$. We also parameterize the inverse dynamics $f_\phi$ with an MLP. For implementation details, please refer to the Appendix B.

**Low-temperature Sampling** In the denoising step of Algorithm 1, we compute $\mu_{k-1}$ and $\Sigma_{k-1}$ from a noisy sequence of states and a predicted noise. We find that sampling $x_{k-1} \sim \mathcal{N}(\mu_{k-1}, \alpha\Sigma_{k-1})$ where the variance is scaled by $\alpha \in [0, 1)$ leads to better quality sequences (corresponding to sampling lower temperature samples). For a proper ablation study, please refer to Appendix C.

## 4 Experiments

In this section, we explore the efficacy of the Decision Diffuser on a variety of decision-making tasks (performance illustrated in Figure 4). In particular, we evaluate **(1)** the ability to recover effective RL policies from offline data, **(2)** the ability to generate behavior that satisfies multiple sets of constraints, **(3)** the ability compose multiple different skills together. In addition, we empirically justify use of classifier-free guidance, low-temperature sampling (Appendix C), and inverse dynamics (Appendix F) and test the robustness of Decision Diffuser to stochastic dynamics (Appendix G).

### 4.1 Offline Reinforcement Learning

**Setup** We first test whether the Decision Diffuser can generate return-maximizing trajectories. To test this, we train a state diffusion process and inverse dynamics model on publicly available

D4RL datasets (Fu et al., 2020). We compare with existing offline RL methods, including model-free algorithms like CQL (Kumar et al., 2020) and IQL (Kostrikov et al., 2022), and model-based algorithms such as trajectory transformer (TT, Janner et al. (2021)) and MoReL (Kidambi et al., 2020). We also compare with sequence-models like the Decision Transformer (DT) (Chen et al. (2021) and diffusion models like Diffuser (Janner et al., 2022).

| Dataset | Environment | BC | CQL | IQL | DT | TT | MOReL | Diffuser | DD |
|---|---|---|---|---|---|---|---|---|---|
| Med-Expert | HalfCheetah | 55.2 | 91.6 | 86.7 | 86.8 | **95** | 53.3 | 79.8 | 90.6 ±1.3 |
| Med-Expert | Hopper | 52.5 | 105.4 | 91.5 | 107.6 | **110.0** | 108.7 | 107.2 | **111.8** ±1.8 |
| Med-Expert | Walker2d | **107.5** | 108.8 | 109.6 | 108.1 | 101.9 | 95.6 | **108.4** | 108.8 ±1.7 |
| Medium | HalfCheetah | 42.6 | 44.0 | 47.4 | 42.6 | 46.9 | 42.1 | 44.2 | **49.1** ±1.0 |
| Medium | Hopper | 52.9 | 58.5 | 66.3 | 67.6 | 61.1 | **95.4** | 58.5 | 79.3 ±3.6 |
| Medium | Walker2d | 75.3 | 72.5 | 78.3 | 74.0 | 79 | 77.8 | 79.7 | **82.5** ±1.4 |
| Med-Replay | HalfCheetah | 36.6 | **45.5** | **44.2** | 36.6 | 41.9 | 40.2 | 42.2 | 39.3 ±4.1 |
| Med-Replay | Hopper | 18.1 | 95 | 94.7 | 82.7 | 91.5 | 93.6 | 96.8 | **100** ±0.7 |
| Med-Replay | Walker2d | 26.0 | 77.2 | 73.9 | 66.6 | **82.6** | 49.8 | 61.2 | 75 ±4.3 |
| **Average** | | 51.9 | 77.6 | 77 | 74.7 | 78.9 | 72.9 | 75.3 | **81.8** |
| Mixed | Kitchen | 51.5 | 52.4 | 51 | - | - | - | - | **65** ±2.8 |
| Partial | Kitchen | 38 | 50.1 | 46.3 | - | - | - | - | **57** ±2.5 |
| **Average** | | 44.8 | 51.2 | 48.7 | - | - | - | - | **61** |

Table 1: **Offline Reinforcement Learning Performance.** We show that Decision Diffuser (DD) either matches or outperforms current offline RL approaches on D4RL tasks in terms of normalized average returns (Fu et al., 2020). We report the mean and the standard error over 5 random seeds.

**Results** Across different offline RL tasks, we find that the Decision Diffuser is either competitive or outperforms many offline RL baselines (Table 1). It also outperforms Diffuser and sequence modeling approaches, such as Decision Transformer and Trajectory Transformer. The difference between Decision Diffuser and other methods becomes even more significant on harder D4RL Kitchen tasks which require long-term credit assignment.

To convey the importance of classifier-free guidance, we also compare with the baseline CondDiffuser, which diffuses over both state and action sequences as in Diffuser without classifier-guidance. In Table 2, we observe that CondDiffuser improves over Diffuser in 2 out of 3 environments. Decision Diffuser further improves over CondDiffuser, performing better across all 3 environments. We conclude that learning the inverse dynamics is a good alternative to diffusing over actions. We further empirically analyze when to use inverse dynamics and when to diffuse over actions in Appendix F. We also compare against CondMLPDiffuser, a policy where the current action is denoised according to a diffusion process conditioned on both the state and return. We see that CondMLPDiffuser performs the worst amongst diffusion models. Till now, we mainly tested on offline RL tasks that have deterministic (or near deterministic) environment dynamics. Hence, we test the robustness of Decision Diffuser to stochastic dynamics and compare it to Diffuser and CQL as we vary the stochasticity in environment dynamics, in Appendix G. Finally, we analyze the runtime characteristics of Decision Diffuser in Appendix E.

## 4.2 CONSTRAINT SATISFACTION

**Setup** We next evaluate how well we can generate trajectories that satisfy a set of constraints using the Kuka Block Stacking environment (Janner et al., 2022) visualized in Figure 5. In this domain, there are four blocks which can be *stacked* as a single tower or *rearranged* into several towers. A constraint like BlockHeight($i$) > BlockHeight($j$) requires that block $i$ be placed above block $j$. We train the Decision Diffuser from $10,000$ expert demonstrations each satisfying one of these constraints. We randomize the positions of these blocks and consider two tasks at inference: sampling trajectories that satisfy a single constraint seen before in the dataset or satisfy a group of constraints for which demonstrations were never provided. In the latter, we ask the Decision Diffuser to generate trajectories so BlockHeight($i$) > BlockHeight($j$) > BlockHeight($k$) for three of the four blocks $i, j, k$. For more details, please refer to Appendix H.

**Results** In both the stacking and rearrangement settings, Decision Diffuser satisfies single constraints with greater success rate than Diffuser (Table 3). We also compare with BCQ (Fujimoto et al., 2019) and CQL (Kumar et al., 2020), but they consistently fail to stack or rearrange the blocks leading to a 0.0 success rate. Unlike these baselines, our method can just as effectively satisfy several constraints together according to Equation 9. For a visualization of these generated trajectories, please see the website https://anuragajay.github.io/decision-diffuser/.

| Hopper−⋆ | Diffuser | CondDiffuser | CondMLPDiffuser | Decision Diffuser |
|---|---|---|---|---|
| Med-Expert | 107.6 | **111.3** | 105.6 | **111.8** ±1.6 |
| Medium | 58.5 | 66.3 | 54.1 | **79.3** ±3.6 |
| Med-Replay | 96.8 | 76.5 | 66.5 | **100** ±0.7 |

Table 2: **Ablations.** Using classifier-free guidance with Diffuser, resulting in `CondDiffuser`, improves performance in 2 (out of 3) environments. Additionally, using inverse dynamics for action prediction in Decision Diffuser improves performance in all 3 environments. `CondMLPDiffuser`, that diffuses over current action given the current state and the target return, doesn't perform as well.

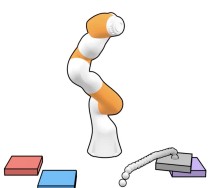

Figure 5: **Kuka Block Stacking task.**

| Environment | Diffuser | DD |
|---|---|---|
| Single Constraint - Stacking | 45.6 ±3.1 | **58.0** ±3.1 |
| Single Constraint - Rearrangement | 58.9 ±3.4 | **62.7** ±3.1 |
| **Single Constraint Average** | 52.3 | **60.4** |
| Multiple Constraints - Stacking | - | **60.3** ±3.1 |
| Multiple Constraints - Rearrangement | - | **67.2** ±3.1 |
| **Multiple Constraints Average** | - | **63.8** |

Table 3: **Block Stacking through Constraint Minimization.** Decision Diffuser (DD) improves over Diffuser in terms of the success rate of generating trajectories satisfying a set of block-stacking constraints. It can also flexibly combine multiple constraints during test time. We report the mean success rate and the standard error over 5 random seeds.

## 4.3 SKILL COMPOSITION

**Setup**  Finally, we look at how to compose different skills together. We consider the Unitree-go-running environment (Margolis & Agrawal, 2022), where a quadruped robot can be found running with various gaits, like bounding, pacing, and trotting. We explore if it is possible to generate trajectories that transition between these gaits after only training on individual gaits. For each gait, we collect a dataset of 2500 demonstrations on which we train Decision Diffuser.

**Results**  During testing, we use the noise model of our reverse diffusion process according to equation 9 to sample trajectories of the quadruped robot with entirely new running behavior. Figure 6 shows a trajectory that begins with bounding but ends with pacing. Appendix I provides additional visualizations of running gaits being composed together. Although it visually appears that trajectories generated with the Decision Diffuser contain more than one gait, we would like to quantify exactly how well different gaits can be composed. To this end, we train a classifier to predict at every time-step or frame in a trajectory the running gait of the quadruped (i.e. bound, pace, or trott). We reuse the demonstrations collected for training the Decision Diffuser to also train this classifier, where our inputs are defined as robot joint states over a fixed period of time (i.e. state sub-sequences of length 10) and the label is the gait demonstrated in this sequence. The complete details of our gait classification procedure can be found in Appendix I.

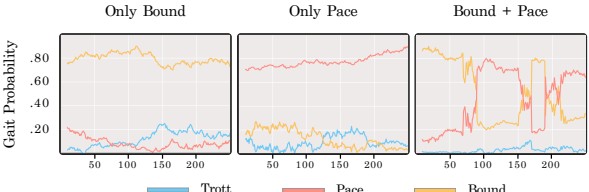

| Condition | Trott | Pace | Bound |
|---|---|---|---|
| Only Bound | 0.8 | 1.0 | 98.2 |
| Only Pace | 1.4 | 97.7 | 0.9 |
| Bound + Pace | 1.4 | 38.5 | 60.1 |

Figure 7: **Classifying Running Gaits.** A classifier predicts the running gait of the quadruped at every timestep. On trajectories generated by conditioning on a single skill, like only bounding or pacing, the classifier predicts the respective gait with largest probability. When conditioned on both skills, some timesteps are classified as bounding while others as pacing.

We use our running gait classifier in two ways: to evaluate how the behavior of the quadruped changes over the course of a single, generated trajectory and to measure how often each gait emerges over several generated trajectories. In the former, we first sample three trajectories from the Decision Diffuser conditioned either on the bounding gait, the pacing gait, or both. For every trajectory, we separately plot the classification probability of each gait over the length of the sequence. As shown in the plots of Figure 7, the classifier predicts bound and pace respectively to be the most likely running gait in trajectories sampled with this condition. When the trajectory is generated by conditioning on both gaits, the classifier transitions between predicting one gait with largest probability to the

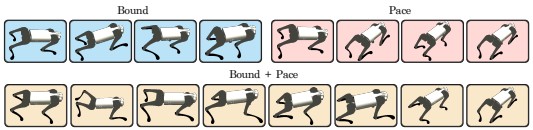

Figure 6: **Composing Movement Skills.** Decision Diffuser can imitate individual running gaits using expert demonstrations and compose multiple different skills together during test time. The results are best illustrated by videos viewable at https://anuragajay.github.io/decision-diffuser/.

other. In fact, there are several instances where the behavior of the quadruped switches between bounding and pacing according to the classifier. This is consistent with the visualizations reported in Figure 6. In the table depicted in Figure 7, we consider 1000 trajectories generated with the Decision Diffuser when conditioned on one or both of the gaits as listed. We record the fraction of time that the quadruped's running gait was classified as either trott, pace, or bound. It turns out that the classifier identifies the behavior as bounding for 38.5% of the time and as pacing for the other 60.1% when trajectories are sampled by composing both gaits. This corroborates the fact that the Decision Diffuser can indeed compose running behaviors despite only being trained on individual gaits.

## 5 RELATED WORK

**Diffusion Models** Diffusion Models is proficient in learning generative models of image and text data (Saharia et al., 2022; Nichol et al., 2021; Nichol & Dhariwal, 2021). It formulates the data sampling process as an iterative denoising procedure (Sohl-Dickstein et al., 2015; Ho et al., 2020). The denoising procedure can be alternatively interpreted as parameterizing the gradients of the data distribution (Song et al., 2021) optimizing the score matching objective (Hyvärinen, 2005) and thus as a Energy-Based Model (Du & Mordatch, 2019; Nijkamp et al., 2019; Grathwohl et al., 2020). To generate data samples (eg: images) conditioned on some additional information (eg:text), prior works (Nichol & Dhariwal, 2021) have learned a classifier to facilitate the conditional sampling. More recent works (Ho & Salimans, 2022) have argued to leverage gradients of an implicit classifier, formed by the difference in score functions of a conditional and an unconditional model, to facilitate conditional sampling. The resulting classifier-free guidance has been shown to generate better conditional samples than classifier-based guidance. Recent works have also used diffusion models to imitate human behavior (Pearce et al., 2023) and to parameterize policy in offline RL (Wang et al., 2022). Janner et al. (2022) generate trajectories consisting of states and actions with an unconditional diffusion model, therefore requiring a trained reward function on noisy state-action pairs. At inference, the estimated reward function guides the reverse diffusion process towards samples of high-return trajectories. In contrast, we do not train reward functions or diffusion processes separately, but rather model the trajectories in our dataset with a single, conditional generative model. This ensures that the sampling procedure of the learned diffusion process is the same at inference as it is during training.

**Reward Conditioned Policies** Prior works (Kumar et al., 2019; Schmidhuber, 2019; Emmons et al., 2021; Chen et al., 2021) have studied learning of reward conditioned policies via reward conditioned behavioral cloning. Chen et al. (2021) used a transformer (Vaswani et al., 2017) to model the reward conditioned policies and obtained a performance competitive with offline RL approaches. Emmons et al. (2021) obtained similar performance as Chen et al. (2021) without using a transformer policy but relied on careful capacity tuning of MLP policy. In contrast, Decision Diffuser can also model constraints or skills and their resulting compositions.

## 6 DISCUSSION

We propose Decision Diffuser, a conditional generative model for sequential decision making. It frames offline sequential decision making as conditional generative modeling and sidesteps the need of reinforcement learning, thereby making the decision making pipeline simpler. By sampling for high returns, it is able to capture the best behaviors in the dataset and outperforms existing offline RL approaches on standard D4RL benchmarks. In addition to returns, it can also be conditioned on constraints or skills and can generate novel behaviors by flexibly combining constraints or composing skills during test time. In this work, we focused on offline sequential decision making, thus circumventing the need for exploration. Using ideas from Zheng et al. (2022), future works could look into online fine-tuning of Decision Diffuser by leveraging entropy of the state-sequence model for exploration. While our work focused on state based environments, it can be extended to image based environments by performing the diffusion in latent space, rather than observation space, as done in Rombach et al. (2022). For a detailed discussion on limitations of Decision Diffuser, please refer to Appendix L.

## ACKNOWLEDGEMENTS

The authors would like to thank Ofir Nachum, Anthony Simeonov and Richard Li for their helpful feedback on an earlier draft of the work; Jay Whang and Ge Yang for discussions on classifier-free guidance; Gabe Margolis for helping with unitree experiments; Micheal Janner for providing visualization code for Kuka block stacking; and the members of Improbable AI Lab for discussions and helpful feedback. We thank MIT Supercloud and the Lincoln Laboratory Supercomputing Center for providing compute resources. This research was supported by an NSF graduate fellowship, a DARPA Machine Common Sense grant, a MURI grant, an MIT-IBM grant, and ARO W911NF-21-1-0097.

This research was also partly sponsored by the United States Air Force Research Laboratory and the United States Air Force Artificial Intelligence Accelerator and was accomplished under Cooperative Agreement Number FA8750-19- 2-1000. The views and conclusions contained in this document are those of the authors and should not be interpreted as representing the official policies, either expressed or implied, of the United States Air Force or the U.S. Government. The U.S. Government is authorized to reproduce and distribute reprints for Government purposes, notwithstanding any copyright notation herein.

## AUTHOR CONTRIBUTIONS

**Anurag Ajay** conceived the framework of viewing decision-making as conditional diffusion generative modeling, implemented the Decision Diffuser algorithm, ran experiments on Offline RL and Skill Composition, and helped in paper writing.

**Yilun Du** helped in conceiving the framework of viewing decision-making as conditional diffusion generative modeling, ran experiments on Constraint Satisfaction, helped in paper writing and advised Anurag.

**Abhi Gupta** helped in running experiments on Offline RL and Skill Composition, participated in research discussions, and played the leading role in paper writing and making figures.

**Joshua Tenenbaum** participated in research discussions.

**Tommi Jaakkola** participated in research discussions and suggested the experiment of classifying running gaits.

**Pulkit Agrawal** was involved in research discussions, suggested experiments related to dynamic programming, provided feedback on writing, positioning of the work, and overall advising.

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

# Appendix

In this appendix, we discuss details of the illustrative examples in Section A. Next, we discuss hyperparameters and architectural details in Section B. We analyze the importance of low temperature sampling in Section C, further explain composition of conditioning variable in Section D, discuss the run-time characteristics of decision diffuser in Section E, discuss when to use inverse dynamics in Section F and analyze robustness of Decision Diffuser to stochastic dynamics in Section G. Finally, we provide details of the Kuka Block Stacking environment in Section H and the Unitree environment in Section I.

## A  ILLUSTRATIVE EXAMPLES

### A.1  IMPLICIT DYNAMIC PROGRAMMING


Training dataset                Generation


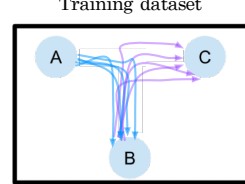 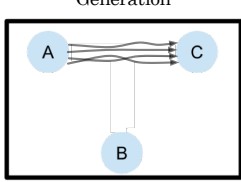

Figure A1: **Illustrative example.** We demonstrate the ability of Decision Diffuser to stitch together sub-optimal trajectories in training dataset to obtain (near) optimal trajectories, thereby implicitly performing dynamic programming in Maze2D-open environment from Fu et al. (2020).

We empirically demonstrate the ability of Decision Diffuser to perform implicit dynamic programming in Maze2D-open environment from Fu et al. (2020). The task in Maze2D-open environment is to reach point C and the reward is negative distance from point C. The training dataset consists of $500$ trajectories from point A to point B and $500$ trajectories from point B to point C. The maximum trajectory length is $50$. During test time, the agent starts from point A and needs to reach point C as quickly as possible. As shown in Figure A1, Decision Diffuser can stitch trajectories in training dataset to form trajectories that goes from point A to point B in (near) straight lines.

### A.2  CONSTRAINT COMBINATION

**Setup**   In linear system robot navigation, Decision Diffuser is trained on $1000$ expert trajectories either satisfying the constraint $\|s_T\| \leq R$ ($R = 1$) or the constraint $\|s_T\| \geq r$ ($r = 0.7$). Here, $s_T = [x_T, y_T]$ represents the final robot state in a trajectory, specifying its final 2d position. The maximum trajectory length is $50$. During test time, Decision Diffuser is asked to generate trajectories satisfying $\|s_T\| \leq R$ and $\|s_T\| \geq r$ to test its ability to satisfy single constraints. Furthermore, Decision Diffuser is also asked to generate trajectories satisfying $r \leq \|s_T\| \leq R$ to test its ability to satisfy combined constraints.

**Results**   Figure 2 shows that Decision Diffuser learns to generate trajectories perfectly (i.e. with $100\%$ success rate) satisfying single constraints in linear system robot navigation. Furthermore, it learns to generate trajectories satisfying the composed constraint in linear system robot navigation with $91.3\%(\pm2.6\%)$ accuracy where the standard error is calculated over 5 random seeds.

## B  HYPERPARAMETER AND ARCHITECTURAL DETAILS

In this section, we describe various architectural and hyperparameter details:

- We represent the noise model $\epsilon_\theta$ with a temporal U-Net (Janner et al., 2022), consisting of a U-Net structure with 6 repeated residual blocks. Each block consisted of two temporal convolutions, each followed by group norm (Wu & He, 2018), and a final Mish nonlinearity (Misra, 2019). Timestep and condition embeddings, both 128-dimensional vectors, are produced by separate 2-layered MLP (with 256 hidden units and Mish nonlinearity) and are concatenated together before getting added to the activations of the first temporal convolution within each block. We borrow the code for temporal U-Net from https://github.com/jannerm/diffuser.

- We represent the inverse dynamics $f_\phi$ with a 2-layered MLP with 512 hidden units and ReLU activations.

- We represent the gait classifier with a 3-layered MLP with 1024 hidden units and ReLU activations.

- We train $\epsilon_\theta$ and $f_\phi$ using the Adam optimizer (Kingma & Ba, 2015) with a learning rate of $2e-4$ and batch size of 32 for $2e6$ train steps.

- We train the gait classifier using the Adam optimizer with a learning rate of $2e-4$ and batch size of 64 for $1e6$ train steps.

- We choose the probability $p$ of removing the conditioning information to be $0.25$.

- We use $K = 100$ diffusion steps.

- We use a planning horizon $H$ of 100 in all the D4RL locomotion tasks, 56 in D4RL kitchen tasks, 128 in Kuka block stacking, 56 in unitree-go-running tasks, 50 in the illustrative example and 60 in Block push tasks.

- We use a guidance scale $s \in \{1.2, 1.4, 1.6, 1.8\}$ but the exact choice varies by task.

- We choose $\alpha = 0.5$ for low temperature sampling.

- We choose context length $C = 20$.

## C  IMPORTANCE OF LOW TEMPERATURE SAMPLING

In Algorithm 1, we compute $\mu_{k-1}$ and $\Sigma_{k-1}$ from a noisy sequence of states and predicted noise. We find that sampling $x_{k-1} \sim \mathcal{N}(\mu_{k-1}, \alpha\Sigma_{k-1})$ (where $\alpha \in [0,1)$) with a reduced variance produces high-likelihood state sequences. We refer to this as low-temperature sampling. To empirically show its importance, we compare performances of Decision Diffuser with different values of $\alpha$ (Table A1). We show that low temperature sampling ($\alpha = 0.5$) gives the best average returns. However, reducing the $\alpha$ to 0 eliminates the entropy in sampling and leads to lower returns. On the other hand, $\alpha = 1.0$ leads to a higher variance in terms of returns of the trajectories.

| Decision Diffuser | Hopper-Medium-Expert |
|---|---|
| $\alpha = 0$ | $104.3 \pm 0.7$ |
| $\alpha = 0.5$ | $\mathbf{111.8} \pm 1.6$ |
| $\alpha = 1.0$ | $107.1 \pm 3.5$ |

Table A1: Low-temperature sampling ($\alpha = 0.5$) allows us to get high return trajectories consistently. While $\alpha = 1.0$ leads to a higher variance in returns of the trajectories, $\alpha = 0.0$ eliminates entropy in the sampling and leads to lower returns.

## D  COMPOSING CONDITIONING VARIABLES

In this section, we detail how Decision Diffuser trained with different conditioning variables $\{\boldsymbol{y}^i(\tau)\}_{i=1}^n$ composes these conditioning variables together. It learns the denoising model $\epsilon_\theta(\boldsymbol{x}_k(\tau), \boldsymbol{y}^i(\tau), k)$ for a given conditioning variable $\boldsymbol{y}^i(\tau)$. From the derivations outlined in prior works (Luo, 2022; Song et al., 2021), we know that $\nabla_{\boldsymbol{x}_k(\tau)} \log q(\boldsymbol{x}_k(\tau)|\boldsymbol{y}^i(\tau)) \propto -\epsilon_\theta(\boldsymbol{x}_k(\tau), \boldsymbol{y}^i(\tau), k)$. Therefore, each conditional trajectory distribution $\{q(\boldsymbol{x}_k(\tau)|\boldsymbol{y}^i(\tau))\}_{i=1}^n$ can be modelled with a single denoising model $\epsilon_\theta$ that conditions on the respective variable $\boldsymbol{y}^i(\tau)$.

In order to compose $n$ different conditioning variables (i.e. skills or constraints), we would like to model $q(\boldsymbol{x}_k(\tau)|\{\boldsymbol{y}^i(\tau)\}_{i=1}^n)$. We assume that $\{\boldsymbol{y}^i(\tau)\}_{i=1}^n$ are conditionally independent given

$\boldsymbol{x}_k(\tau)$. Thus, we can factorize as follows:

$$q(\boldsymbol{x}_k(\tau)|\{\boldsymbol{y}^i(\tau)\}_{i=1}^n) \propto q(\boldsymbol{x}_k(\tau)) \prod_{i=1}^n \frac{q(\boldsymbol{x}_k(\tau)|\boldsymbol{y}^i(\tau))}{q(\boldsymbol{x}_k(\tau))} \quad \text{(Bayes Rule)}$$

$$\Rightarrow \log q(\boldsymbol{x}_k(\tau)|\{\boldsymbol{y}^i(\tau)\}_{i=1}^n) \propto \log q(\boldsymbol{x}_k(\tau)) + \sum_{i=1}^n (\log q(\boldsymbol{x}_k(\tau)|\boldsymbol{y}^i(\tau)) - \log q(\boldsymbol{x}_k(\tau)))$$

$$\Rightarrow \nabla_{\boldsymbol{x}_k(\tau)} \log q(\boldsymbol{x}_k(\tau)|\{\boldsymbol{y}^i(\tau)\}_{i=1}^n) = \nabla_{\boldsymbol{x}_k(\tau)} \log q(\boldsymbol{x}_k(\tau))$$
$$+ \sum_{i=1}^n (\nabla_{\boldsymbol{x}_k(\tau)} \log q(\boldsymbol{x}_k(\tau)|\boldsymbol{y}^i(\tau)) - \nabla_{\boldsymbol{x}_k(\tau)} \log q(\boldsymbol{x}_k(\tau)))$$

$$\Rightarrow \epsilon_\theta(\boldsymbol{x}_k(\tau), \{\boldsymbol{y}^i(\tau)\}_{i=1}^n, k) = \epsilon_\theta(\boldsymbol{x}_k(\tau), \varnothing, k) + \sum_{i=1}^n (\epsilon_\theta(\boldsymbol{x}_k(\tau), \boldsymbol{y}^i(\tau), k) - \epsilon_\theta(\boldsymbol{x}_k(\tau), \varnothing, k))$$

Using the above equations, we can sample from $q(\boldsymbol{x}_0(\tau)|\{\boldsymbol{y}^i(\tau)\}_{i=1}^n)$ with classifier free guidance using the perturbed noise:

$$\hat{\epsilon} := \epsilon_\theta(\boldsymbol{x}_k(\tau), \varnothing, k) + \omega(\epsilon_\theta(\boldsymbol{x}_k(\tau), \{\boldsymbol{y}^i(\tau)\}_{i=1}^n, k) - \epsilon_\theta(\boldsymbol{x}_k(\tau), \varnothing, k))$$
$$= \epsilon_\theta(\boldsymbol{x}_k(\tau), \varnothing, k) + \omega \sum_{i=1}^n (\epsilon_\theta(\boldsymbol{x}_k(\tau), \boldsymbol{y}^i(\tau), k) - \epsilon_\theta(\boldsymbol{x}_k(\tau), \varnothing, k))$$

We use the perturbed noise to compose skills or combine constraints at test time. This derivation was borrowed from Liu et al. (2022) and is presented here for completeness.

While the composition of conditioning variables $\{\boldsymbol{y}^i(\tau)\}_{i=1}^n$ requires them to be conditionally independent given the state trajectory $\boldsymbol{x}_0(\tau)$, we empirically observe that this condition doesn't have to be strictly satisfied. However, we require composition of conditioning variables to be feasible (i.e. $\exists \, \boldsymbol{x}_0(\tau)$ that satisfies all the conditioning variables). When the composition is infeasible, Decision Diffuser produces trajectories with incoherent behavior, as expected. This is best illustrated by videos viewable at https://anuragajay.github.io/decision-diffuser/.

**Requirements on the dataset** First, the dataset should have a diverse set of demonstrations that shows different ways of satisfying each conditioning variable $\boldsymbol{y}^i(\tau)$. This would allow Decision Diffuser to learn diverse ways of satisfying each conditioning variable $\boldsymbol{y}^i(\tau)$. Since we use inverse dynamics to extract actions from the predicted state trajectory $\boldsymbol{x}_0(\tau)$, we assume that the state trajectory $\boldsymbol{x}_0(\tau)$ resulting from the composition of different conditioning variables contains consecutive state pairs $(s_t, s_{t+1})$ that come from the same distribution that generated the demonstration dataset. Otherwise, inverse dynamics can give erroneous predictions.

# E   RUNTIME CHARACTERISTIC OF DECISION DIFFUSER

We analyze the runtime characteristics of Decision Diffuser in this section. After training the Decision Diffuser on trajectories from the D4RL Hopper-Medium-Expert dataset, we plan in the corresponding environment according to Algorithm 1. Every action taken in the environment requires running 100 reverse diffusion steps to generate a state sequence taking on average 1.26s in wall-clock time. We can improve the run-time of planning by warm-starting the state diffusion as suggested in Janner et al. (2022). Here, we start with a generated state sequence (from the previous environment step), run forward diffusion for a fixed number of steps, and finally run the same number of reverse diffusion steps from the partially noised state sequence to generate another state sequence. Warm-starting in this way allows us to decrease the number of denoising steps to 40 (0.48s on average) without any loss in performance, to 20 (0.21s on average) with minimal loss in performance, and to 5 with less than 20% loss in performance (0.06s on average). We demonstrate the trade-off between performance, measured by normalized average return achieved in the environment, and planning time, measured in wall-clock time after warm-starting the reverse diffusion process, in Figure A2.

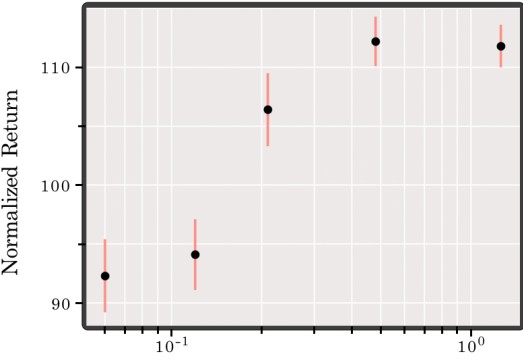

Performance vs. Run–time

Figure A2: **Performance vs planning time.** We visualize the trade-off between performance, measured by normalized average return achieved in the environment, and planning time, measured in wall-clock time after warm-starting the reverse diffusion process.

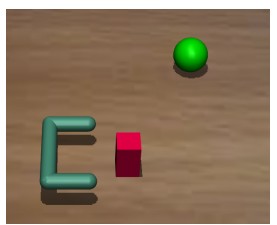

Figure A3: **Block push environment.**

| Environment | BC | CondDiffuser | Decision Diffuser |
|---|---|---|---|
| Position Control | 57.3 ±1.2 | **87.3** ±3.1 | **87.8** ±2.8 |
| Torque Control | 55.2 ±1.5 | 71.8 ±3.4 | **84.7** ±2.2 |

Table A2: **Block pushing with different controls.** Decision Diffuser and CondDiffuser perform similarly when the agent uses position control. However, when the agent uses torque control, CondDiffuser performs worse than Decision Diffuser given it's harder to diffuse over non-smooth action trajectories. We use the success rate of the red cube reaching the green circle as the performance metric. We report the mean success rate and the standard error over 5 random seeds.

## F    WHEN TO USE INVERSE DYNAMICS?

In this section, we try to analyze further when using inverse dynamics is better than diffusing over actions. Table 2 showed that Decision Diffuser outperformed CondDiffuser on 3 hopper environment, thereby suggesting that inverse dynamics is a better alternative to diffusing over actions. Our intuition was that sequences over actions, represented as joint torques in our environments, tend to be more high-frequency and less smooth, thus making it harder for the diffusion model to predict (Kingma et al., 2021). We now try to verify this intuition empirically.

**Setup** We choose Block Push environment adapted from Gupta et al. (2018) where the goal is to push the red cube to the green circle. When the red cube reaches the green circle, the agent gets a reward of +1. The state space is 10-dimensional consisting of joint angles (3) and velocities (3) of the gripper, COM of the gripper (2) and position of the red cube (2). The green circle's position is fixed and at an initial distance of $0.5$ from COM of the gripper. The red cube (of size $0.03$) is initially at a distance of $0.1$ from COM of the gripper and at an angle $\theta$ sampled from $\mathcal{U}(-\pi/4, \pi/4)$ at the start of every episode. The task horizon is 60 timesteps.

There are 2 control types: (i) torque control, where the agent needs to specify joint torques (3 dimensional) and (ii) position control where the agent needs to specify the position change of COM of the gripper and the angular change in gripper's orientation $(\Delta x, \Delta y, \Delta\phi)$ (3 dimensional). While action trajectories from position control are smooth, the action trajectories from torque control have higher frequency components.

**Offline dataset collection** To collect the offline data, we use Soft Actor-Critic (SAC) (Haarnoja et al., 2018) first to train an expert policy for 1 million environment steps. We then use 1 million environment transitions as our offline dataset, which contains expert trajectories collected towards the end of the training and random action trajectories collected at the beginning of the training. We collect 2 datasets, one for each control type.

**Results** Table A2 shows that Decision Diffuser and CondDiffuser perform similarly when the agent uses position control. This is because action trajectories resulting from position control are smoother and hence easier to model with diffusion. However, when the agent uses torque control, CondDiffuser performs worse than Decision Diffuser, given the action trajectories have higher frequency components and hence are harder to model with diffusion.

## G  ROBUSTNESS TO STOCHASTIC DYNAMICS

| p | BC | Decision Diffuser | Diffuser | CQL |
|---|---|---|---|---|
| 0.00 | $55.2 \pm 1.5$ | **$84.7$**$\pm 2.2$ | $72.4 \pm 1.4$ | $73.2 \pm 2.3$ |
| 0.05 | $49.3 \pm 3.6$ | **$77.3$**$\pm 3.1$ | $63.2 \pm 2.9$ | $61.8 \pm 3.7$ |
| 0.10 | $25.8 \pm 3.8$ | **$53.2$**$\pm 4.1$ | $52.3 \pm 4.6$ | $51.2 \pm 4.3$ |
| 0.15 | $15.1 \pm 4.3$ | **$41.3$**$\pm 4.9$ | **$41.6$**$\pm 5.1$ | $42.2 \pm 5.5$ |

Table A3: **Robustness to stochastic dynamics.** Decision Diffuser's performance suffers when stochasticity is introduced in dynamics function. While it still outperforms Diffuser and CQL when $p = 0.05$, its performance becomes similar to that of Diffuser and CQL for higher $p$ values. We use the success rate of the red cube reaching the green circle as the performance metric. We report the mean success rate and the standard error over 5 random seeds.

We empirically analyze robustness of Decision Diffuser to stochasticity in dynamics function.

**Setup**  We use Block Push environment, described in Appendix F, with torque control. However, we inject stochasticity into the environment dynamics. For every environment step, we either sample a random action from $\mathcal{U}([-1, -1, -1], [1, 1, 1])$ with probability $p$ or execute the action given by the policy with probability $(1 - p)$. We use $p \in \{0, 0.05, 0.1, 0.15\}$ in our experiments.

**Offline dataset collection**  We collect separate offline datasets for different block push environments, each characterized by a different value of $p$. Each offline dataset consists of 1 million environment transitions collected using the method described in Appendix F.

**Results**  Table A3 characterizes how the performance of BC, Decision Diffuser, Diffuser, and CQL changes with increasing stochasticity in the environment dynamics. We observe that the Decision Diffuser outperforms Diffuser and CQL for $p = 0.05$, however all methods including the Decision Diffuser settle to a similar performance for larger values of $p$.

Several works (Paster et al., 2022; Yang et al., 2022) have shown that the performance of return-conditioned policies suffers as the stochasticity in environment dynamics increases. This is because the return-conditioned policies aren't able to distinguish between high returns from good actions and high returns from environment stochasticity. Hence, these return-conditioned policies can learn sub-optimal actions that got associated with high-return trajectories in the dataset due to environment stochasticity. Given Decision diffuser uses return conditioning to generate actions in offline RL, its performance also suffers when stochasticity in environment dynamics increases.

Some recent works (Yang et al., 2022; Villaflor et al., 2022) address the above issue by learning a latent model for future states and then conditioning the policy on predicted latent future states rather than returns. Conditioning Decision Diffuser on future state information, rather than returns, would make it more robust to stochastic dynamics and could be an interesting avenue for future works.

## H  KUKA BLOCK STACKING

In the Kuka blocking stacking environment, the underlying goal is to stack a set of blocks on top of each other. Models have trained on a set of demonstration data, where a set of 4 blocks are sequentially stacked on top of each other to form a block tower.

We construct state-space plans of length 128. Following (Janner et al., 2022), we utilize a close-loop controller to generate actions for each state in our state-space plan (controlling the 7 degrees of freedom in joints). The total maximum trajectory length plan in Kuka block stacking is 384. We detail differences between the two consider conditional stacking environments below:

- **Stacking** In the stacking environment, at test time we wish to again construct a tower of four blocks.

- **Rearrangement** In the rearrangement environment, at test time wish to stack blocks in a configuration where a set of blocks are above a second set. This set of stack-place relations may not precisely correspond to a single block tower (can instead construct two block towers), making this environment an out-of-distribution challenge.

In addition to Diffuser (Janner et al., 2022), we used goal-conditioned variants of `CQL` (Kumar et al., 2020) and `BCQ` (Fujimoto et al., 2019) as baselines for the block stacking and rearrangement with single constraint. However, they get a success rate of 0.0.

# I   UNITREE GO RUNNING

We consider Unitree-go-running environment (Margolis & Agrawal, 2022) where a quadruped robot runs in 3 different gaits: bounding, pacing, and trotting. The state space is 56 dimensional, the action space is 12 dimensional, and the maximum trajectory length is 250.

As described in Section 4.3, we train Decision Diffuser on expert trajectories demonstrating individual gaits. During testing, we compose the noise model of our reverse diffusion process according to equation 9. This allows us to sample trajectories of the quadruped robot with entirely new running behavior. Figures A4,A5,A6 shows the ability of Decision Diffuser to imitate bounding, trotting and pacing and their combinations.

## I.1   QUANTITATIVE VERIFICATION OF COMPOSITION

We now try to quantitatively verify whether the trajectories resulting from composition of 2 gaits does indeed contain only those 2 gaits.

**Setup**  We learn a gait classifier that takes in a sub-sequence of states (of length 10) and predicts the gait-ID. It is represented by a 3-layered MLP with 1024 hidden units and ReLU activations that concatenates the sub-sequence of states (of length 10) into a single vector of dimension 560 before taking it in as an input. We train the gait classifier on the demonstration dataset. To ensure that the learned classifier can predict gait-ID on trajectories generated by the composition of skills, we use MixUp-style (Zhang et al., 2017) data augmentation during training. We create a synthetic sub-sequence of length 10 by concatenating two sampled sub-sequence (from the demonstration dataset) of length $l_i$ and $l_j$ (where $l_i + l_j = 10$) from gaits with ID $i$ and $j$ and give it a label $\frac{l_i}{l_i+l_j}$one-hot$(i) +$ $\frac{l_j}{l_i+l_j}$one-hot$(j)$. During training, we sample a sub-sequence from the demonstration dataset with 70% probability and a sythenthic sub-sequence with 30% probability. We train the classifier for $2e6$ train steps with a learning rate of $2e - 4$ and a batch size of 64.

**Results**  Figures A4,A5,A6 show that the classifier's prediction is consistent with the visualized composed trajectories. Furthermore, we use Decision diffuser to act in the environment and generate 1000 trott trajectories, 1000 pace trajectories, 1000 bound trajectories, and 1000 composed trajectories for each possible pair of individual gaits. We then evaluate the learned gait classifier on these trajectories and compute the percentage of timesteps a particular gait has the highest probability. From Figures A4,A5,A6, we can see that if trajectories are generated by the composition of two gaits, then those two gaits will have the two highest probabilities across different timesteps in those trajectories.

## I.2   A SIMPLE BASELINE FOR COMPOSITION

Let one-hot$(i)$ and one-hot$(j)$ represent two different gaits that can be generated using noise models $\epsilon_\theta(\boldsymbol{x}_k(\tau), \text{one-hot}(i), k)$ and $\epsilon_\theta(\boldsymbol{x}_k(\tau), \text{one-hot}(j), k)$ respectively. To compose these gaits, we compose the above-mentioned noise models using equation 9. As an alternative, we see if the noise model $\epsilon_\theta(\boldsymbol{x}_k(\tau), \text{one-hot}(i) + \text{one-hot}(j), k)$ can lead to composed gaits. However, we observe that $\epsilon_\theta(\boldsymbol{x}_k(\tau), \text{one-hot}(i) + \text{one-hot}(j), k)$ catastrophically fail to generate any gait (see videos at https://anuragajay.github.io/decision-diffuser/). This happens because the condition variable one-hot$(i) +$ one-hot$(j)$ was never seen by the noise model $\epsilon_\theta$ during training.

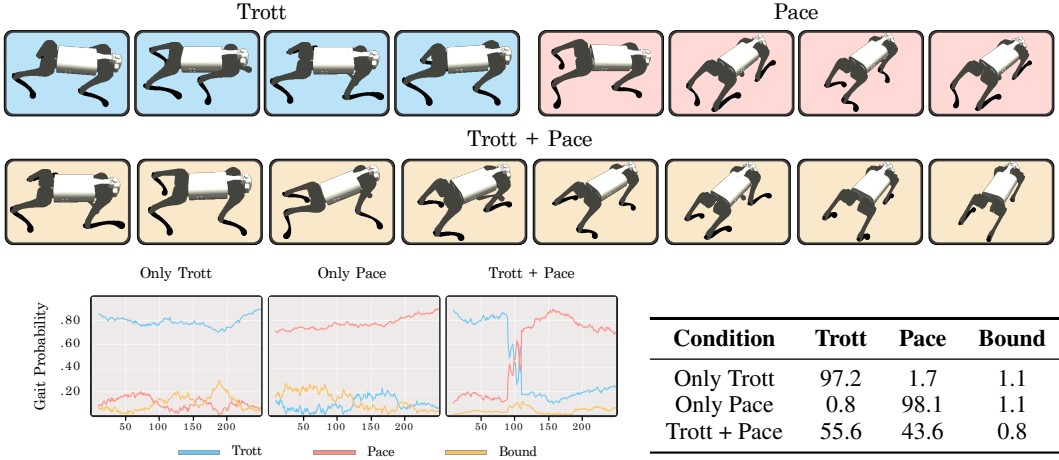

Figure A4: **Composing Trott and Pace.** Decision Diffuser can imitate individual running gaits using expert demonstrations and compose multiple different skills together during test time. The results are best illustrated by videos viewable at https://anuragajay.github.io/decision-diffuser/.

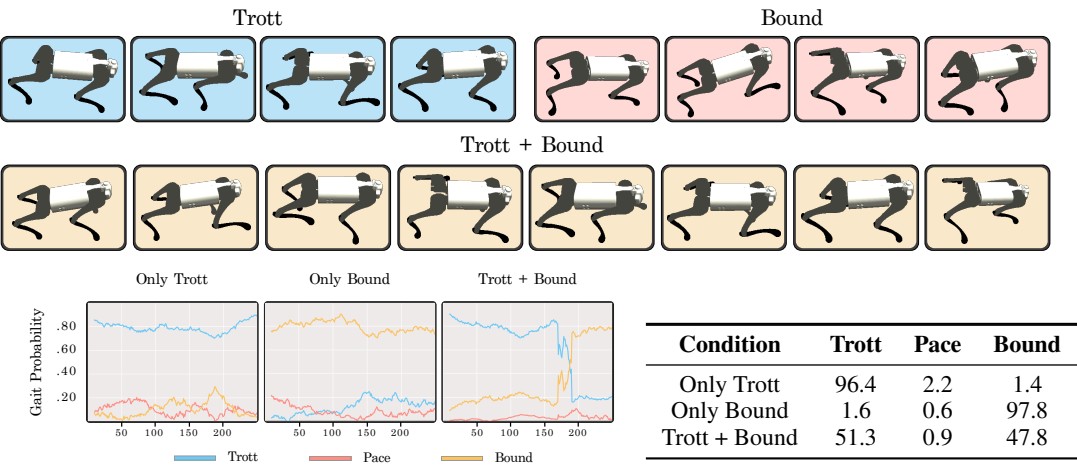

Figure A5: **Composing Trott and Bound.** Decision Diffuser can imitate individual running gaits using expert demonstrations and compose multiple different skills together during test time. The results are best illustrated by videos viewable at https://anuragajay.github.io/decision-diffuser/.

## J NOT COMPOSITIONS WITH DECISION DIFFUSER

Decision diffuser can also support "NOT" composition. Suppose we wanted to sample from $q(\boldsymbol{x}_0(\tau)|\text{NOT } \boldsymbol{y}^j(\tau))$. Let $\{\boldsymbol{y}^i(\tau)\}_{i=1}^n$ be the set of all conditioning variables. Then, following derivations from Liu et al. (2022) and using $\beta = 1$, we can sample from $q(\boldsymbol{x}_0(\tau)|\text{NOT } \boldsymbol{y}^j(\tau))$ using the perturbed noise:

$$\hat{\epsilon} := \epsilon_\theta(\boldsymbol{x}_k(\tau), \varnothing, k) + \omega(\sum_{i \neq j}(\epsilon_\theta(\boldsymbol{x}_k(\tau), \boldsymbol{y}^i(\tau), k) - \epsilon_\theta(\boldsymbol{x}_k(\tau), \varnothing, k))$$

$$- (\epsilon_\theta(\boldsymbol{x}_k(\tau), \boldsymbol{y}^j(\tau), k) - \epsilon_\theta(\boldsymbol{x}_k(\tau), \varnothing, k)))$$

We demonstrate the ability of Decision Diffuser to support "NOT" composition by using it to satisfy constraint of type `BlockHeight(i) > BlockHeight(j)` **AND** (**NOT** `BlockHeight(j) > BlockHeight(i)`) in Kuka block stacking task, as visualized in videos at https://anuragajay.github.io/decision-diffuser/. As the Decision Diffuser does not provide an explicit density estimate for each skill, it can't natively support OR composition.

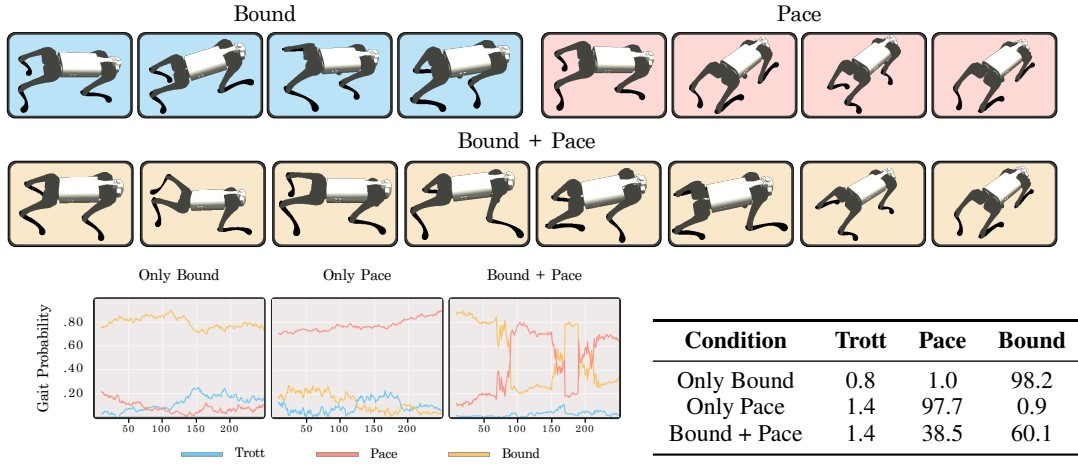

Figure A6: **Composing Bound and Pace.** Decision Diffuser can imitate individual running gaits using expert demonstrations and compose multiple different skills together during test time. The results are best illustrated by videos viewable at https://anuragajay.github.io/decision-diffuser/.

## K    COMPARING Q-FUNCTION GUIDED DIFFUSION AND CLASSIFIER-FREE GUIDED DIFFUSION

Classifier-free guided diffusion and Q-value guided diffusion are theoretically equivalent. However, as noted in several works (Nichol et al., 2021; Ho & Salimans, 2022; Saharia et al., 2022), classifier-free guidance performs better than classifier guidance (i.e. Q function guidance in our case) in practice. This is due to following reasons:

- Classifier-guided diffusion models learns an unconditional diffusion model along with a classifier (Q-function in our case) and uses gradients from the classifier to perform conditional sampling. However, the unconditional diffusion model doesn't need to focus on conditional modeling during training and only cares about conditional generation during testing after it has been trained. In contrast, classifier-free guidance relies on conditional diffusion model to estimate gradients of the implicit classifier. Since the conditional diffusion model, learned when using classifier-free guidance, focuses on conditional modeling during train time, it performs better in conditional generation during test time.

- Q function trained on an offline dataset can erroneously predict high Q values for out-of-distribution actions given any state. This problem has been extensively studied in offline RL literature (Kumar et al., 2020; Fujimoto et al., 2019; Levine et al., 2020). In online RL, this issue is automatically corrected when the policy acts in the environment, thinking an action to be good but then receives a low reward for it. In offline RL, this issue can't be corrected easily; hence, the learned Q-function can often guide the diffusion model towards out-of-distribution actions that might be sub-optimal. In contrast, classifier-free guidance circumvents the issue of learning a Q-function and directly conditions the diffusion model on returns. Hence, classifier-free guidance doesn't suffer due to errors in learned Q-functions and hence performs better than Q-function guided diffusion.

## L    LIMITATIONS OF DECISION DIFFUSER

We summarize the limitations of Decision Diffuser:

- **No partial observability** Decision Diffuser works with fully observable MDPs. Naive extensions to partially observed MDPs (POMDPs) may cause *self-delusions* (Ortega et al., 2021) in Decision Diffuser. Hence, extending Decision Diffuser to POMDPs could be an exciting avenue for future work.

- **Inability to explore the environment and update itself in online setting** In this work, we focused on offline sequential decision making, thus circumventing the need for exploration. Using ideas from Zheng et al. (2022), future works could look into online fine-tuning of Decision Diffuser by leveraging entropy of the state-sequence model for exploration.

- **Experiments on only state-based environments** While our work focused on state based environments, it can be extended to image based environments by performing the diffusion in latent space, rather than observation space, as done in Rombach et al. (2022).

- **Only AND and NOT compositions are supported** Since Decision Diffuser does not provide an explicit density estimate for each condition variable, it can't natively support OR composition.

- **Performance degradation in environments with stochastic dynamics** In environments with highly stochastic dynamics, Decision Diffuser loses its advantage and performs similarly to Diffuser and CQL. To tackle environments with stochastic dynamics, recent works (Yang et al., 2022; Villaflor et al., 2022) propose learning a latent model for future states and then conditioning the policy on predicted latent future states rather than returns. Conditioning Decision Diffuser on future state information, rather than returns, would make it more robust to stochastic dynamics and could be an interesting avenue for future works.

- **Performance in limited data regime** Since diffusion models are prone to overfitting in case of limited data, Decision Diffuser is also prone to overfitting in limited data regime.

