# OpenReview forum: "Is Conditional Generative Modeling all you need for Decision Making?"
_ICLR.cc/2023/Conference — ICLR 2023 notable top 5%_

### Official Review · Reviewer_WRRm · 2022-10-24

**Confidence:** 3
**Correctness:** 4
**Technical Novelty And Significance:** 4
**Empirical Novelty And Significance:** 3
**Recommendation:** 6

**Clarity, Quality, Novelty And Reproducibility:**

The paper is of high quality, clarity, and novelty. Since I'm not an expert in refinement learning/decision-making. I'm not entirely sure about the reproducibility.

**Strength And Weaknesses:**

## Strength
- Modeling decision-making as the conditional generative model is novel and also interesting to the community.
- Via conditional generative process, the method shows the ability to compose skills in order to maximize the rewards. This is impressive.
- The paper in general is of high quality in terms of supportive experiments, and motivation.

## Weakness
I'm not an expert in decision-making/Reinforcement learning. So I'm not entirely sure if I follow the problem definition. To be specific, what's the design choice that leads to the ability to compose skills? And do other reinforcement learning-based method have the same ability? It would be better to make it simpler.

**Summary Of The Paper:**

The paper presents the decision diffuser for decision-making. The core idea is to model the decision making as a conditional generative model, which is different from the complex reinforcement learning-based works. Specifically, the method models a policy as a conditional generative model where the diffusion process is applied over states and gets return-maximizing trajectories. The experimental results show the performance on par with recent offline RL-based methods.

**Summary Of The Review:**

From my perspective, the paper is of high quality. It presents a well-motivated story.
- Good motivation for modeling decision-making as a generative model.
- More importantly, the method shows the ability to compose skills, which impresses me a lot.
- The experimental results are also impressive. Though the conditional generative model is simple, it shows the performance on par with other complex reinforcement learning.

Although I'm not very clear about some details of the diffusion process in the composition of skills, I tend to accept this paper given the impressive results.

---

> ### Author Response · Authors · 2022-11-15
> **Author's Response (1/2)**
>
> We thank reviewer WRRm for evaluating our work. We now answer the following concerns raised in the review.
>
> >What design choices lead to the ability of composing skills?
>
> We have described how decision-making can be formulated as a standard problem in conditional generative modelling. Prior work like Decision Transformer [1] and Trajectory Transformer [2] have specifically modelled trajectories with transformers. In this work, we make the key design choice of leveraging Diffusion Probabilistic Models [3] as a way to model the trajectory distribution instead. As shown in our paper, this not only leads to superior return-conditional performance but also provides a means for conditioning beyond returns, like constraints or skills. We would like to clarify that the procedure for composing more than one conditioning variable is more generally a property of any diffusion process [4] and not specific to the Decision Diffuser. We describe the details below for completeness:
>
> Decision Diffuser learns the denoising model $\\epsilon_\\theta(x_t(\\tau), y^i(\\tau), t)$ for a given conditioning variable $y^i(\\tau)$. From the derivations outlined in [5] or the score-based interpretation of diffusion models [6], we know that $\\nabla_{x_t(\\tau)} \\log q(x_t(\\tau)|y^i(\\tau)) \\propto -\\epsilon_\\theta(x_t(\\tau), y^i(\\tau), t)$. Therefore, each conditional trajectory distribution $\\{q(x_0(\\tau)|y^i(\\tau))\\}_{i=1}^n$ can be modelled with a single denoising model $\\epsilon_\\theta$ that conditions on the respective variable $y^i(\\tau)$.
>
> In order to compose $n$ different conditioning variables (i.e. skills or constraints), we would like to model $q(x\_0(\\tau)| (y^i(\\tau))\_{i=1}^n)$. We assume that $(y^i(\\tau))\_{i=1}^n$ are conditionally independent given $x\_0(\\tau)$. Thus, we can factorize as follows:
>
> $$
> q(x\_0(\tau)|(y^i(\tau))\_{i=1}^n) \propto q(x\_0(\tau)) \prod\_{i=1}^n \frac{q(x\_0(\tau)|y^i(\tau))}{q(x_0(\tau))}
> $$ (Bayes Rule)
>
> $$
> \Rightarrow \log q(x\_0(\tau)|(y^i(\tau))\_{i=1}^n) \propto \log q(x\_0(\tau)) + \sum\_{i=1}^n (\log q(x\_0(\tau)|y^i(\tau)) - \log q(x\_0(\tau)))
> $$
>
>  $$
> \Rightarrow \nabla_{x\_t(\tau)} \log q(x\_0(\tau)|(y^i(\tau))\_{i=1}^n) = \nabla\_{x\_t(\tau)} \log q(x\_0(\tau)) + \sum
> \_{i=1}^n (\nabla\_{x\_t(\tau)} \log q(x\_0(\tau)|y^i(\tau)) - \nabla\_{x\_t(\tau)} \log q(x\_0(\tau)))
> $$
>
> $$
> \Rightarrow \epsilon\_\theta(x\_t(\tau), (y^i(\tau))\_{i=1}^n, t) = \epsilon\_\theta(x\_t(\tau), \emptyset, t) + \sum\_{i=1}^n (\epsilon\_\theta(x\_t(\tau), y^i(\tau), t) - \epsilon\_\theta(x\_t(\tau), \emptyset, t))
> $$
>
> For this reason, we can sample from $q(x_0(\tau)|(y^i(\tau))_{i=1}^n)$ with classifier-free guidance according to the perturbed noise:
>
> $\hat{\epsilon} \coloneqq \epsilon_\theta(x_t(\tau), \emptyset, t) + s(\epsilon_\theta(x_t(\tau), (y^i(\tau))_{i=1}^n, t) - \epsilon_\theta(x_t(\tau), \emptyset, t))$
>
> $= \epsilon_\theta(x_t(\tau), \emptyset, t) + s\sum_{i=1}^n (\epsilon_\theta(x_t(\tau), y^i(\tau), t) - \epsilon_\theta(x_t(\tau), \emptyset, t))$
>
> In summary, modelling trajectories with a diffusion process allows us to make use of the result above to compose several conditioning variables together. We specifically use the perturbed noise above to compose skills or combine constraints at test time. We detail this explanation in Appendix D of the updated draft.
>
> >And do other reinforcement learning-based method have the same ability?
>
> Prior work has proposed Soft Q-learning, an RL algorithm, as a way to compose different kinds of behavior. Consider each constraint or skill as defining a task with reward function $r_{i}(s,a)$. According to [7], a policy that optimizes for all of the tasks in the set $\mathcal{C}$ with reward $\frac{1}{|\mathcal{C}|}\sum\_{j \in \mathcal{C}}{r\_{j}(s,a)}$ can be obtained from acting greedy with respect to the $Q$-function $\frac{1}{|\\mathcal{C}|}\sum\_{j \in \mathcal{C}}{\hat{Q}\_j(s,a)}$, where $\hat{Q}\_{j}(s,a)$ is the Q-function for the policy that solves task $j$. Composing behaviors in this way requires learning a $Q$-function for each task through interaction with the environment. The Decision Diffuser in contrast composes behaviors from offline datasets without knowledge of task-specific rewards or learning any kind of $Q$-function.

---

> > ### Author Response · Authors · 2022-11-15
> > **Author's Response (2/2)**
> >
> > **References**
> > 1. Decision Transformer: Reinforcement Learning via Sequence Modeling. Lili Chen^, Kevin Lu^, Aravind Rajeswaran, Kimin Lee, Aditya Grover, Michael Laskin, Pieter Abbeel, Aravind Srinivas+, Igor Mordatch+. NeurIPS 2021 (^ Equal contribution, + Equal advising).
> > 2. Offline Reinforcement Learning as One Big Sequence Modeling Problem. Michael Janner, Qiyang Li, Sergey Levine. NeurIPS 2021.
> > 3. Denoising Diffusion Probabilistic Models. Jonathan Ho, Ajay Jain, Pieter Abbeel. arXiv:2006.11239.
> > 4. Compositional Visual Generation with Composable Diffusion Models. Nan Liu, Shuang Li, Yilun Du, Antonio Torralba, Joshua B. Tenenbaum. ECCV 2022.
> > 5. Understanding Diffusion Models: A Unified Perspective. Calvin Luo. arXiv:2208.11970.
> > 6. Denoising Diffusion Implicit Models. Jiaming Song, Chenlin Meng, Stefano Ermon. ICLR 2021.
> > 7. Composable Deep Reinforcement Learning for Robotic Manipulation. Tuomas Haarnoja, Vitchyr Pong, Aurick Zhou, Murtaza Dalal, Pieter Abbeel, Sergey Levine. ICRA 2018.

---

> ### Author Response · Authors · 2022-11-17
> **Looking forward to further discussions!**
>
> Dear Reviewer,
>
> Thank you for your time and effort in reviewing our work. We have provided detailed clarification to address the issues raised in your comments. If our response has addressed your concerns, we would be grateful if you could re-evaluate our work.
>
> If you have any additional questions or comments, we would be happy to have further discussions.
>
> Thanks,
>
> Authors

---

### Official Review · Reviewer_pyhf · 2022-10-25

**Confidence:** 3
**Correctness:** 4
**Technical Novelty And Significance:** 3
**Empirical Novelty And Significance:** 3
**Recommendation:** 8

**Clarity, Quality, Novelty And Reproducibility:**

The paper is well written and organized. The proposed method is evaluated empirically across a broad range of offline reinforcement learning scenarios for return maximization. Skill composition and constraint satisfaction are also explored.

Diffusion models are proposed to be used for sequential trajectory data in Jammer et al. (2022), where both states and actions are modeled jointly. The model proposed in this paper separates these two and uses an inverse dynamics model to model actions. Furthermore, they adopt a classifier free guidance approach (as opposed to classifier guidance in Jammer et al., 2022). These incremental advancements yield superior results in offline reinforcement learning settings.

Janner, M., Du, Y., Tenenbaum, J.B. and Levine, S., 2022. Planning with Diffusion for Flexible Behavior Synthesis. arXiv preprint arXiv:2205.09991.

**Strength And Weaknesses:**

Strengths:
- The paper is well written. Diffusion modeling for planning is an exciting and timely contribution given that diffusion models are beating state of the art in various other tasks in synthesis.
- The proposed technique is shown to tackle compositionality of skills and constraints at inference time, which is interesting.

Weaknesses:
- One limitation of diffusion models is that the generations are slow. It would be great if authors can comment on the runtime characteristics of the proposed model.
- It would be good to discuss limitations of the model. Some questions I have:
  * How are the out-of-distribution generalization characteristics of the model?
  * Can the model capture variable length trajectories?
  * I am intrigued by the skill composition idea. Currently only “AND” style composition is demonstrated. What other compositions does this method can support?
  * How does the method perform with limited data?

Other comments:
- The loss function used to train the model combines losses which are used to train the diffusion model and inverse dynamics models equally. What implications does equal weighting have?
- The paper relies on diffusion models and class conditioning capabilities of such models (i.e. classifier free guidance) but I found it odd that the modeling choice is not mentioned in the abstract.
- What is the performance metric used in Figure 4? Similarly, Table 3 is lacking an explanation of the metric used.


**Summary Of The Paper:**

This paper proposes Decision Diffuser, a diffusion-based model for sequential decision making where only the states of a trajectory are modeled and an inverse dynamics model is used to predict actions. Classifier free guidance is used to bring in conditional information, in the form of maximizing returns, satisfying constraints or composing together skills. This is used with low-temperature sampling at inference time yielding improved results. The proposed method is evaluated to generate trajectories for return maximization with offline reinforcement learning datasets. Kuka Block stacking and Unitree-go-running environments are used to assess the performance of generated trajectories in constraint satisfaction and skill composition tasks, respectively.


**Summary Of The Review:**

The paper explores diffusion models for planning, which is a timely contribution since diffusion models are beating state of the art in various problems. It offers incremental advancements over the state of the art in this domain however the proposed advancements are shown to yield superior performance on various tasks. I think this is a good paper and I recommend acceptance.

---

> ### Author Response · Authors · 2022-11-15
> **Author's Response (1/2)**
>
> We thank reviewer pyhf for thoroughly evaluating our work - it has helped us improve our paper. We now answer the following concerns raised in the review:
>
> >One limitation of diffusion models is that the generations are slow. It would be great if authors can comment on the runtime characteristics of the proposed model.
>
> We agree that sampling according to a diffusion process can indeed be quite slow. We appreciate the reviewer for encouraging us to look further into the run-time characteristics of the Decision Diffuser. After training the Decision Diffuser on trajectories from the D4RL Hopper-Medium-Expert dataset, we plan in the corresponding environment according to Algorithm 1. For every action taken in the environment, this requires running 100 reverse diffusion steps in order to generate a state sequence taking on average 1.26s in wall-clock time. We can improve the run-time of planning by warm-starting the state diffusion as suggested in [4]. Here, we start with a generated state sequence (from the previous environment step), run forward diffusion for a fixed number of steps, and finally run the same number of reverse diffusion steps from the partially noised state sequence to generate another state sequence. Warm-starting in this way allows us to decrease the number of denoising steps to 40 (0.48s on average) without any loss in performance, to 20 (0.21s on average) with minimal loss in performance, and to 5 with less than 20% loss in performance (0.06s on average). We demonstrate the trade-off between performance, measured by normalized average return (taken from [1]) achieved in the environment, and planning time, measured in wall-clock time after warm-starting the reverse diffusion process, through Figure A1 in Appendix E of the updated draft.
>
> >The loss function used to train the model combines losses which are used to train the diffusion model and inverse dynamics models equally. What implications does equal weighting have?
>
> The loss used to train the Decision Diffuser is highlighted below:
>
> $$
> \mathcal{L}(\theta, \phi) \coloneqq E\_{t, \tau\in\mathcal{D}, \beta\sim Bern(p)}[||\epsilon - \epsilon\_{\theta}(\boldsymbol{x}\_{t}(\tau), (1-\beta)y(\tau) + \beta\emptyset, t)||^{2}] + E\_{(s, a, s') \in \mathcal{D}}[||a-f\_{\phi}(s, s')||^2]
> $$
>
> Notice that the state diffusion process is modeled by parameters $\theta$ and the inverse dynamics is modeled by parameters $\phi$. Since each term in our loss contains $\theta$ or $\phi$, but not both, it is equivalent to training the denoising model $\epsilon_{\theta}$ and inverse dynamics $f_{\phi}$ separately with the same dataset. For this reason, the two terms can be weighted arbitrarily.
>
> >The paper relies on diffusion models and class conditioning capabilities of such models (i.e. classifier free guidance) but I found it odd that the modeling choice is not mentioned in the abstract.
>
> We agree with the reviewer--a key design choice that makes skill and constraint composition possible is that we use diffusion processes as our conditional generative model. We have updated our abstract to highlight this modeling choice.
>
> >What is the performance metric used in Figure 4? Similarly, Table 3 is lacking an explanation of the metric used.
>
> In Figure 4, we compare the performance of Decision Diffuser (DD) to Conservative Q-learning (CQL) and Behavior Cloning (BC) on three benchmarks: D4RL Locomotion, D4RL Kitchen, and Kuka Block Stacking. Each plot demonstrates the performance of these methods on one of the three benchmarks. The performance metric shown on the y-axis depends on the plot. The plot comparing these methods on the D4RL Locomotion and Kitchen benchmark uses normalized average returns (taken from [1]) as the performance metric. In order to measure performance on the Kuka Block Stacking benchmark, we use the success rate of how often trajectories generated by one of the methods (i.e. DD, CQL, BC) satisfy the block stacking constraints of the given task.
>
> We have added a description of the aforementioned performance metrics in Figure 4. We reuse the performance metric for evaluating Decision Diffuser on the Kuka Block Stacking task in Table 3, where we additionally compare to Diffuser.

---

> > ### Author Response · Authors · 2022-11-15
> > **Author's response (2/2)**
> >
> > >How are the out-of-distribution generalization characteristics of the model?
> >
> > Decision Diffuser learns the denoising model $\epsilon\_\theta(x\_t(\tau), y^i(\tau), t)$ for a given conditioning variable $y^i(\tau)$. From the derivations outlined in [2] or the score-based interpretation of diffusion models [3], we know that $\nabla_{x\_t(\tau)} \log q(x\_t(\tau)|y^i(\tau)) \propto -\epsilon\_\theta(x\_t(\tau), y^i(\tau), t)$. Therefore, each conditional trajectory distribution $(q(x\_0(\tau)|y^i(\tau)))\_{i=1}^n$ can be modelled with a single denoising model $\epsilon\_\theta$ that conditions on the respective variable $y^i(\tau)$.
> >
> > If those variables $y^i(\tau)$ are conditionally independent given the state trajectory $x\_t(\tau)$, the score function of the composed conditional state trajectory distribution $\nabla\_{x\_t(\tau)} \log q(x\_0(\tau)|(y^i(\tau))\_{i=1}^n)$ can be written in terms of score functions of individual conditional trajectory distribution $\nabla_{x_t(\tau)} \log q(x_t(\tau)|y^i(\tau))$, each represented by the denoising model $\epsilon\_\theta$. Therefore, Decision diffuser can approximate composed conditional state trajectory distribution by modeling its score function $\nabla\_{x\_t(\tau)} \log q(x\_0(\tau)|(y^i(\tau))\_{i=1}^n)$ in terms of $\epsilon\_\theta$ even though Decision Diffuser hasn't been trained on the composed distribution. We describe this in detail in Appendix D of the updated draft.
> >
> > >Can the model capture variable length trajectories?
> >
> > Our state diffusion process does not assume a fixed planning horizon. This means that the denoising model $\epsilon\_{\theta}(x_t(\tau), y(\tau), t)$ can denoise trajectories $\tau$ of arbitrary length. As described in our work, we define $x\_t \coloneqq (s\_k, s\_{k+1}, ..., s\_{k+H-1})\_t$ as a noisy sequence of $H$ states represented as a two-dimensional array where the height is the dimension of a single state and the width is $H$.  In order to generate a trajectory with the Decision Diffuser, we first sample $x\_T \sim \mathcal{N}(0,I)$ and then iteratively denoise $x\_T$ with our model $\epsilon\_{\theta}$, which also outputs a two-dimensional array with the same size as $x\_T$. Since we choose to parameterize $\epsilon\_{\theta}$ with a temporal U-Net architecture, a neural network consisting of repeated convolutional residual blocks, the width of  $\epsilon\_{\theta}(x_{t}(\tau), y(\tau), t)$ depends only on the width of the input $x_{t}(\tau)$ as often the case with CNNs. For this reason, the Decision Diffuser can denoise trajectories of length other than $H$ without requiring any modifications.
> >
> > >I am intrigued by the skill composition idea. Currently only “AND” style composition is demonstrated. What other compositions does this method can support?
> >
> > Decision diffuser can also support "NOT" composition. Suppose we wanted to sample from $q(x\_0(\tau)| NOT(y^j(\tau)))$. Let $(y^i(\tau))\_{i=1}^n$ be the set of all conditioning variables. Then, following derivations from [5] and using $\beta=1$, we can sample from $q(x\_0(\tau)|NOT(y^j(\tau)))$ using the perturbed noise:
> >
> > $$
> > \hat{\epsilon} \coloneqq \epsilon\_\theta(x\_t(\tau),\emptyset, t) + s (\sum\_{i \neq j} (\epsilon\_\theta(x\_t(\tau),y^i(\tau), t) - \epsilon_\theta(x\_t(\tau), \emptyset, t)) - (\epsilon\_\theta(x\_t(\tau),y^j(\tau), t) - \epsilon\_\theta(x\_t(\tau), \emptyset, t)))
> > $$
> >
> > We demonstrate the ability of Decision Diffuser to support "NOT" composition by using it to satisfy constraint of type "BlockHeight(i)>BlockHeight(j) AND (NOT BlockHeight(j) > BlockHeight(l))" in Kuka block stacking task, as visualized in [https://sites.google.com/view/decisiondiffuser/](https://sites.google.com/view/decisiondiffuser/). As the Decision Diffuser does not provide an explicit density estimate for each skill, it can't natively support OR composition. We added this discussion in Appendix J of the updated draft.
> >
> > >How does the method perform with limited data?
> >
> > Since diffusion models are prone to overfitting in case of limited data, Decision Diffuser is also prone to overfitting in limited data regime.
> >
> > **References**
> > 1. D4RL: Datasets for Deep Data-Driven Reinforcement Learning. Justin Fu, Aviral Kumar, Ofir Nachum, George Tucker, Sergey Levine. arXiv:2004.07219.
> > 2. Understanding Diffusion Models: A Unified Perspective. Calvin Luo. arXiv:2208.11970.
> > 3. Denoising Diffusion Implicit Models. Jiaming Song, Chenlin Meng, Stefano Ermon. ICLR 2021.
> > 4. Planning with Diffusion for Flexible Behavior Synthesis. Michael Janner^, Yilun Du^, Joshua Tenenbaum, and Sergey Levine. ICML 2022 (^ Equal contribution).
> > 5. Compositional Visual Generation with Composable Diffusion Models. Nan Liu, Shuang Li, Yilun Du, Antonio Torralba, Joshua B. Tenenbaum. ECCV 2022.

---

> ### Author Response · Authors · 2022-11-17
> **Looking forward to further discussions!**
>
> Dear Reviewer,
>
> Thank you for your time and effort in reviewing our work. We have provided detailed clarification and additional experiments to address the issues raised in your comments. If our response has addressed your concerns, we would be grateful if you could re-evaluate our work.
>
> If you have any additional questions or comments, we would be happy to have further discussions.
>
> Thanks,
>
> Authors

---

### Official Review · Reviewer_9svn · 2022-10-25

**Confidence:** 3
**Correctness:** 3
**Technical Novelty And Significance:** 2
**Empirical Novelty And Significance:** 2
**Recommendation:** 8

**Clarity, Quality, Novelty And Reproducibility:**

The paper is very well written, perhaps a slightly more expanded discussion and detailed comparison against previously proposed methods such as the Diffuser could be helpful to clearly and compactly state the novelty. The quality of the experiments shown is high, including comparison against strong SOTA methods. What I would have liked to see for an even higher quality paper is some critical discussion and analysis (via ablations or control experiments) of the main innovations (see Improvements). The method is a fairly incremental improvement of e.g. the Diffuser and the idea of using generative modeling for decision making has by now been widely explored - yet I think the paper has sufficient novelty. Sufficient details for reproducibility of the work seem present in the appendix.

**Strength And Weaknesses:**

**Main contributions, Impact**
1) Use of classifier-free guided diffusion. Classifier-guided diffusion requires a Q-value estimation procedure such that Q-values can be used to guide diffusion (which has been explored in Janner et al. 2022). The current work avoids this via classifier-free guidance which requires low-temperature sampling in datasets where demonstrations are of mixed quality. Empirically, this leads to improvements over previous methods - with a grain of salt it seems to be easier to pick out the high performing trajectories than learning good Q-value estimates. Impact: low to medium - it remains unclear under what exact conditions (both theoretically and empirically) classifier-free guidance / low-temperature sampling outperforms Q-value guidance and why.

2) Diffusion of state-trajectories only, rather than state-action trajectories (which is compared to in the paper and shown to perform worse). This comes at the cost of requiring a good inverse dynamics model. Current impact: low - similar to 1) the empirical results are in favor of the version proposed in the paper but it remains unclear what the practical cost of requiring the inverse dynamics model is; in particular how robust the method is against errors and imperfections in the inverse dynamics estimate (is performance very brittle or quite robust). Under what conditions is it better to rely on an approximate inverse dynamics model, and when is it better to diffuse state-action trajectories?

3) The paper shows how to incorporate combinations of constraints and skills: by identifying subsets of training trajectories corresponding to an individual constraint/skill and augmenting them with a label that can later be used to condition on. The approach is fairly straightforward, but interesting. Impact: low - the approach is currently only minimally explored in the paper and further conceptual/theoretical characterization of the kinds of constraints and the corresponding requirements for the dataset would be needed for higher impact.

**Strengths**
 *  Very well written paper
 *  Empirical results show benefits of proposed method compared to high-quality SOTA methods on standard benchmarks
 * Incorporating constraints and combining skills via constraints is interesting.

**Weaknesses**
 *  Often the paper’s main aim seems to be to reach the minimal result sufficient for publication, which is mainly to achieve good benchmark results. While this is a valid strategy to avoid risks, it also creates a sense of lack of ambition and depth. I personally think there are some very interesting ideas presented in the paper but they often seems to be addressed in a minimal sense. Beyond the empirical comparisons, a strong characterization of the advantages and disadvantages of these ideas is missing; and I would be very excited to see such a discussion and analysis (both empirically and theoretically). I think it would help raise the significance of each of the main contributions mentioned above.
 * To make the point above more concrete:
   * It remains unclear when precisely and why classifier-free guidance and state-prediction only is preferable - when and why is it beneficial to avoid estimating Q-values at the cost of requiring an inverse dynamics model (how robust is each approach to errors in the approximate Q-values/dynamics, what are the conditions w.r.t. the dataset that make low-temperature sampling a good choice, etc).
   * It remains unclear what exact conditions are needed for constraints/skills to be “combinable” (see comment under improvements).
 * The current experiments on combining skills are a bit hard to interpret - while they lead to somewhat visually different gaits it is unclear whether these should be considered successful combinations of skills. What’s missing is a clearly stated and (quantitatively) measurable goal for combining skills (which is well defined in the case of combining constraints).

**Improvements**
1) Naively it seems that the approach introduced allows to combine individual constraints / skills, as long as there is a separate subset of training trajectories for each constraint/skill and the combination at test time is a conjunction of constraints such that the intersection of the corresponding sets of trajectories constitutes a set of valid solutions (which corresponds to the situation in Fig 1). Is this a hard condition? Can something theoretical be said about the allowed combinations of individual constraints (conjunction, disjunction, exclusive disjunction)? What happens theoretically and empirically if satisfying a combination of constraints requires novel trajectories that have no support in the training data (i.e. such that simply “filtering” out the right training trajectories via conditioning and some mild generalization is not sufficient)?

2) Combining skills needs some quantitative goal/metric, and an empirical evaluation. The videos are interesting and promising, but it is unclear how to interpret the results, let alone how other methods could be compared in terms of performance.

3) Perturbation analysis comparing the Diffuser and Decision Diffuser in terms of robustness to noise or errors in the Q-value estimates/inverse-dynamics model respectively. While theoretical results would be nice, empirical results are probably easier to obtain and might already be insightful. Related: how well do both methods handle quite noisy dynamics? Since the latter two requires running additional experiments, potentially even designing toy environments, I do not have a hard expectation to see these results after the rebuttal; but they would certainly make the paper stronger.

4) Discussion of limitations should be expanded
 * What are restrictions/requirements on the dataset, particularly w.r.t. allowing for combinable constraints/skills?
 * The current method is limited to fully observable MDPs, partial observability and latent variables might pose a hard to overcome problem, with self-delusions [1] potentially being a fundamental obstacle).
 * It seems unlikely that the method can be used to improve significantly beyond the performance of demonstrations (unlike RL, at least in theory).

**Minor comments**

A) What is the relation between constraint conditioning as presented in the paper and more general task conditioning (via natural language prompts) as in e.g. DeepMind’s Gato?

B) Sec. 3 “It is useful to solve RL from offline data, both without relying on TD-learning and without risking distribution-shift.” - what are the conditions/requirements for the latter to be possible (at least informally)? In order to not *risk distribution shift* strong conditions on training-data coverage seem required.

C) What is the fundamental difference between constraints and skills - are they more than simply two different names to refer to subsets of trajectories in the training data?

D) “In contrast to these works, in addition to modeling returns, Decision Diffuser can also model constraints or skills and generate novel behaviors by flexibly combining multiple constraints or skills during test time.” - while this has not been explored in previous publications, is there a fundamental problem that prevents previous methods from using the same approach as used in this paper (i.e. augmenting data with a one-hot indicator for constraints/skills, and conditioning on that during generation)?

E) P4, typo: “choices fof diffusion”

[1] Shaking the foundations: delusions in sequence models for interaction and control, Ortega et al. 2021


**Summary Of The Paper:**

**Update after rebuttal** I am very happy to see a very extensive and detailed authors' rebuttal with a number of additional evaluations and results. Most of my questions have been answered and my criticism has been addressed to a sufficient degree. Though there could be some detailed follow-up discussions, I think overall the paper is now ready for acceptance - I am on the fence between a 6 and an 8, but am very slightly leaning towards the latter, so I will raise my score from 5 to 8.

The paper explores the use of a diffusion model to generate reward-conditioned state-trajectories to reach goals and satisfy constraints in MDPs given a large set of state-action-reward trajectories (solving decision-making via conditional generative modeling). In contrast to previous work on synthesizing high-expected-return policies from expert data, this paper uses a classifier-free guided diffusion model. Additionally, the paper proposes to diffuse over state-trajectories only, rather than state-reward trajectories; which is empirically shown to lead better policies but requires learning an inverse dynamics model. The method is compared against a number of state-of-the-art competitors on two standard benchmarks (D4RL tasks, and Kuka Block stacking tasks). Finally, the paper shows how conditional generation of trajectories can not only be used for conditioning on high-rewards but also other subsets of training trajectories which either satisfy certain constraints or belong to a certain type of skill (such as a particular gait for running).

**Summary Of The Review:**

The paper makes some incremental, but sensible improvements to previously proposed work - and the empirical results justify the modifications. While the main ingredients that went into the improvements have been reported before, the particular combination is novel. To me personally, the paper could be more impactful and of higher quality by justifying the introduced changes beyond mere overall performance on some benchmarks and discussing advantages and disadvantages in detail (including some insight into when and why the introduced modifications are beneficial compared to the previous method). Another novelty introduced is the possibility to combine constraints/skills - this is an interesting idea, but currently it is explored rather superficially, a deeper exploration (as suggested earlier in my review) would make the work stronger. Overall I am quite on the fence for this paper - while there is a clear and sensible main idea and empirical results are in favor of the idea, the paper also has a bit of a flavor of “we tried something and it worked, but we didn’t really look into it any further”. I think that’s potentially a missed chance, since the paper could be quite strong with a bit more work and attempting to answer some of the harder questions. I am therefore currently voting for a borderline rejection, and would not be upset if the paper got accepted - but to me personally the paper is currently on  the level of a very well written and very strong workshop contribution; promising main results but a bit of a lack in depth and clear and well-understood main findings. I am happy to reconsider my final verdict in light of the other reviews and author discussion.

---

> ### Author Response · Authors · 2022-11-15
> **Author's response (1/6)**
>
> We thank reviewer 9svn for thoroughly evaluating our work - it has helped us improve our paper. We now answer the following concerns raised in the review.
>
> > It remains unclear under what exact conditions (both theoretically and empirically) classifier-free guidance / low-temperature sampling outperforms Q-value guidance and why.
>
> We would like to clarify that classifier-free guidance and Q-value guidance are theoretically equivalent. However, as noted in several works [1,2,3], classifier-free guidance performs better than classifier guidance (i.e. Q function guidance in our case) in practice. This is due to following reasons:
> - Classifier-guided diffusion models learns an unconditional diffusion model along with a classifier (Q-function in our case) and uses gradients from the classifier to perform conditional sampling. However, the unconditional diffusion model doesn't need to focus on conditional modeling during training and only cares about conditional generation during testing after it has been trained. In contrast, classifier-free guidance relies on conditional diffusion model to estimate gradients of the implicit classifier. Since the conditional diffusion model, learned when using classifier-free guidance, focuses on conditional modeling during train time, it performs better in conditional generation during test time.
> - Q function trained on an offline dataset can erroneously predict high Q values for out-of-distribution actions given any state. This problem has been extensively studied in offline RL literature [4,5,6,7]. In online RL, this issue is automatically corrected when the policy acts in the environment thinking an action to be good but then receives a low reward for it. In offline RL, this issue can't be corrected easily; hence, the learned Q-function can often guide the diffusion model towards out-of-distribution actions that might be sub-optimal. In contrast, classifier-free guidance circumvents the issue of learning a Q-function and directly conditions the diffusion model on returns. Hence, classifier-free guidance doesn't suffer due to errors in learned Q-functions and hence performs better than Q-function guided diffusion.
>
> We have added this discussion in Appendix K of the updated draft.
>
> > Under what conditions is it better to rely on an approximate inverse dynamics model, and when is it better to diffuse state-action trajectories?
>
> Table 2 in the paper showed that Decision Diffuser outperformed CondDiffuser on $3$ hopper environment, thereby suggesting that inverse dynamics is a better alternative to diffusing over actions. Our intuition was that sequences over actions, which were represented as joint torques in our environments, tend to be more high-frequency and less smooth, thus making it harder for the diffusion model to predict (as noted in [8]). We now try to verify this intuition empirically.
>
> **Setup** We choose Block Push environment adapted from [9] where the goal is to push the red cube to the green circle. When the red cube reaches the green circle, the agent gets a reward of +1. The state space is $10$-dimensional consisting of joint angles ($3$) and velocities ($3$) of the gripper, COM of the gripper ($2$) and position of the red cube ($2$). The green circle's position is fixed and at an initial distance of $0.5$ from COM of the gripper. The red cube (of size $0.03$) is initially at a distance of $0.1$ from COM of the gripper and at an angle $\theta$ sampled from $\mathcal{U}(-\pi/4, \pi/4)$ at the start of every episode. The task horizon is $60$ timesteps.
>
> There are 2 control types: (i) torque control, where the agent needs to specify joint torques ($3$ dimensional) and (ii) position control where the agent needs to specify the position change of COM of the gripper and the angular change in gripper's orientation $(\Delta x, \Delta y, \Delta \phi)$  ($3$ dimensional). While action trajectories from position control are smooth, the action trajectories from torque control have higher frequency components.
>
> **Offline dataset collection** To collect the offline data, we use Soft Actor-Critic (SAC) [10] first to train an expert policy for $1$ million environment steps. We then use $1$ million environment transitions as our offline dataset, which contains expert trajectories collected towards the end of the training and random action trajectories collected at the beginning of the training. We collect 2 datasets, one for each control type.
>
> **Results** Table A2 in the paper shows that Decision Diffuser and CondDiffuser perform similarly when the agent uses position control. This is because action trajectories resulting from position control are smoother and hence easier to model with diffusion. However, when the agent uses torque control, CondDiffuser performs worse than Decision Diffuser, given the action trajectories have higher frequency components and hence are harder to model with diffusion.
>
> We have included the above discussion in Appendix F of the updated draft.

---

> > ### Author Response · Authors · 2022-11-15
> > **Author's response (2/6)**
> >
> > >Further conceptual/theoretical characterization of the kinds of constraints and the corresponding requirements for the dataset would be needed for higher impact.
> > >What are restrictions/requirements on the dataset, particularly w.r.t. allowing for combinable constraints/skills?
> >
> > For $n$ different conditioning variables $(y^i(\tau))\_{i=1}^n$ (i.e. skills or constraints) to be composable, we assume that $(y^i(\tau))\_{i=1}^n$are conditionally independent given $x\_0(\tau)$. This assumption is required for the derivation of Equation 9 in the paper:
> >
> > $$
> > \hat{\epsilon} = \epsilon\_\theta(x\_t(\tau), \emptyset, t) + s\sum_{i=1}^n (\epsilon\_\theta(x\_t(\tau), y^i(\tau), t) - \epsilon\_\theta(x\_t(\tau), \emptyset, t))
> > $$
> >
> > We show this in Appendix D of the updated draft.
> >
> > While the composition of conditioning variables $(y^i(\tau))\_{i=1}^n$ requires them to be conditionally independent given the state trajectory $x\_0(\tau)$, we empirically observe that this condition doesn't have to be strictly satisfied. However, we require composition of conditioning variables to be feasible (i.e. $\exists\ x_0(\tau)$ that satisfies all the conditioning variables). When the composition is infeasible, Decision Diffuser produces trajectories with incoherent behavior, as expected. This is best illustrated by videos viewable at [https://sites.google.com/view/decisiondiffuser/](https://sites.google.com/view/decisiondiffuser/).
> >
> > **Requirements on the dataset**: First, the dataset should have a diverse set of demonstrations that shows different ways of executing each skill or satisfying each constraint. This would allow Decision Diffuser to learn diverse ways of performing each skill or satisfying each constraint. Since we use inverse dynamics to extract actions from the predicted state trajectory $x\_0(\tau)$, we assume that the state trajectory $x\_0(\tau)$ resulting from the composition of different skills (or combination of different constraints) contains consecutive state pairs $(s\_t, s\_{t+1})$ that come from the same distribution that generated the demonstration dataset. Otherwise, inverse dynamics can give erroneous predictions.
> >
> > We have also added this discussion in Appendix D of the updated draft.
> >
> > >a strong characterization of the advantages and disadvantages of these ideas is missing; and I would be very excited to see such a discussion and analysis (both empirically and theoretically). Discussion of limitations should be expanded: (i) What are restrictions/requirements on the dataset, particularly w.r.t. allowing for combinable constraints/skills? (ii) The current method is limited to fully observable MDPs, partial observability and latent variables might pose a hard to overcome problem, with self-delusions potentially being a fundamental obstacle). (iii) It seems unlikely that the method can be used to improve significantly beyond the performance of demonstrations (unlike RL, at least in theory).
> > 1. We have included this discussion in Appendix D of the updated draft. We have also explained the requirements on the dataset in one of our previous answers.
> > 2. We agree that our method only applies to MDPs.
> > 3. We train Decision diffuser on offline RL datasets containing sub-optimal trajectories ({HalfCheetah, Hopper, Walker2d}-medium and {HalfCheetah, Hopper, Walker2d}-medium-replay, Kitchen-mixed, Kitchen-partial) and then test it in corresponding environments (Section 4.1). It outperforms behavioral cloning (BC) in all these environments, thereby improving beyond the behavioral policy. Furthermore, on average, Decision Diffuser outperforms SOTA offline RL approaches on these D4RL datasets. Hence, it often gives more improvements than traditional RL approaches trained on offline datasets.
> >
> >   Decision diffuser doesn't have the ability to further explore an environment and improve by getting feedback from the environment in an online fashion. Using ideas from [11], future works could look into online fine-tuning of Decision Diffuser by leveraging entropy of the state-sequence models for exploration.
> >
> > We summarize the discussion of limitations of Decision Diffuser in Appendix L of the updated draft.

---

> > > ### Author Response · Authors · 2022-11-15
> > > **Author's Response (3/6)**
> > >
> > > >Combining skills needs some quantitative goal/metric, and an empirical evaluation. The videos are interesting and promising, but it is unclear how to interpret the results, let alone how other methods could be compared in terms of performance.
> > >
> > > To quantify exactly how well different gaits can be composed, we train a classifier to predict at every time-step or frame in a trajectory the running gait of the quadruped (i.e. bound, pace, or trott). We reuse the demonstrations collected for training the Decision Diffuser to also train this classifier, where our inputs are defined as robot joint states over a fixed period of time (i.e. state sub-sequences of length $10$) and the label is the gait demonstrated in this sequence. The complete details of our gait classification procedure can be found in Appendix I of the updated draft.
> > >
> > > We use our running gait classifier in two ways: to evaluate how the behavior of the quadruped changes over the course of a single, generated trajectory and to measure how often each gait emerges over several generated trajectories. In the former, we first sample three trajectories from the Decision Diffuser conditioned either on the bounding gait, the pacing gait, or both. For every trajectory, we separately plot the classification probability of each gait over the length of the sequence. As shown in the plots of Figure 7, the classifier predicts bound and pace respectively to be the most likely running gait in trajectories sampled with this condition. When the trajectory is generated by conditioning on both gaits, the classifier transitions between predicting one gait with largest probability to the other. In fact, there are several instances where the behavior of the quadruped switches between bounding and pacing according to the classifier. This is consistent with the visualizations reported in Figure 6. In the table depicted in  Figure 7, we consider $1000$ trajectories generated with the Decision Diffuser when conditioned on one or both of the gaits as listed. We record the fraction of time that the quadruped's running gait was classified as either trott, pace, or bound. It turns out that the classifier identifies the behavior as bounding for 38.5% of the time and as pacing for the other 60.1% when trajectories are sampled by composing both gaits. This corroborates the fact that the Decision Diffuser can indeed compose running behaviors despite only being trained on individual gaits. This discussion has been included in Section 4.3 of the updated draft.

---

> > > > ### Author Response · Authors · 2022-11-15
> > > > **Author's response (4/6)**
> > > >
> > > > >Perturbation analysis comparing the Diffuser and Decision Diffuser in terms of robustness to noise or errors in the Q-value estimates/inverse-dynamics model respectively. While theoretical results would be nice, empirical results are probably easier to obtain and might already be insightful. Related: how well do both methods handle quite noisy dynamics? Since the latter two requires running additional experiments, potentially even designing toy environments, I do not have a hard expectation to see these results after the rebuttal; but they would certainly make the paper stronger.
> > > >
> > > > We empirically analyze robustness of Decision Diffuser to stochasticity in dynamics function.
> > > >
> > > > **Setup** We use Block Push environment, described in Appendix F (of the updated draft), with torque control. However, we inject stochasticity into the environment dynamics. For every environment step, we either sample a random action from $\mathcal{U}([-1,-1,-1], [1,1,1])$ with probability $p$ or execute the action given by the policy with probability $(1-p)$. We use $p \in \{0, 0.05, 0.1, 0.15\}$ in our experiments.
> > > >
> > > > **Offline dataset collection** We collect separate offline datasets for different block push environments, each characterized by a different value of $p$. Each offline dataset consists of 1 million environment transitions collected using the method described in Appendix F of the updated draft.
> > > >
> > > > **Results** Table A3 characterizes how the performance of BC, Decision Diffuser, Diffuser, and CQL changes with increasing stochasticity in the environment dynamics. We observe that the Decision Diffuser outperforms Diffuser and CQL for $p=0.05$, however all methods including the Decision Diffuser settle to a similar performance for larger values of $p$.
> > > >
> > > > Several works [12,13] have shown that the performance of return-conditioned policies suffers as the stochasticity in environment dynamics increases. This is because the return-conditioned policies aren't able to distinguish between high returns from good actions and high returns from environment stochasticity. Hence, these return-conditioned policies can learn sub-optimal actions that get associated with high-return trajectories in the dataset due to environment stochasticity. Given Decision diffuser uses return conditioning to generate actions in offline RL, its performance also suffers when stochasticity in environment dynamics increases.
> > > >
> > > > Some recent works [12,14] address the above issue by learning a latent model for future states and then conditioning the policy on predicted latent future states rather than returns. Conditioning Decision Diffuser on future state information, rather than returns, would make it more robust to stochastic dynamics and could be an interesting avenue for future work.
> > > >
> > > > This discussion has been included in Appendix G of the updated draft.
> > > >
> > > > >What is the relation between constraint conditioning as presented in the paper and more general task conditioning (via natural language prompts) as in e.g. DeepMind’s Gato?
> > > >
> > > > Constraint conditioning is just another type of conditioning where constraints are presented as one-hot indicators. We show that we can solve for a new constraint that hasn’t been seen before as long as we can express the new constraint as a combination of already-seen constraints. We further detail this in Section 3.3, Appendix D and Appendix J.
> > > >
> > > > >What is the fundamental difference between constraints and skills - are they more than simply two different names to refer to subsets of trajectories in the training data?
> > > >
> > > > Constraints are defined by the final state in our case, but skills are defined by the attainment of some behavior. For eg: In Kuka block stacking, the constraint BlockHeight(i)>BlockHeight(j) is satisfied if block i is placed higher than block j in final state of the task. Similarly, in unitree go environment, a particular gait, say trott, is attained if the quadruped is moving forward with a trott (i.e. diagonal pairs of legs move forward at the same time with a moment of suspension between each beat).

---

> > > > > ### Author Response · Authors · 2022-11-15
> > > > > **Author's response (5/6)**
> > > > >
> > > > > >“In contrast to these works, in addition to modeling returns, Decision Diffuser can also model constraints or skills and generate novel behaviors by flexibly combining multiple constraints or skills during test time.” - while this has not been explored in previous publications, is there a fundamental problem that prevents previous methods from using the same approach as used in this paper (i.e. augmenting data with a one-hot indicator for constraints/skills, and conditioning on that during generation)?
> > > > >
> > > > > In Kuka block stacking, Decision diffuser is required to generate trajectories that satisfy the given constraint. While Decision diffuser is only trained on trajectories satisfying a single constraint, it needs to generate trajectories satisfying a set of constraints during the test time. It's able to do that by using equation 9 in the paper to compose noise model $\epsilon_\theta$ conditioned on different single constraints contained in the given set of constraints. While prior works [15,16] have the ability to imitate single constraint, it's unclear how they will combine multiple constraints together given they only see single constraints during training and they don't learn a score function.
> > > > >
> > > > > For skill composition, Decision Diffuser learns to imitate individual quadruped gaits as well compose those gaits during test-time. Given those gaits are represented as one-hot vectors, let trott be [1,0,0], pace be [0,1,0] and bound be [0,0,1]. One alternate way to compose two gaits without using Equation 9 is to condition the noise model of Decision Diffuser on sum of one-hot vectors of those individual gaits. For eg: if we want to compose bound and pace, we can try conditioning the noise model on [0,1,1]. However, this conditioning catastrophically fails as Decision Diffuser has never seen the gait ID [0,1,1] during training (see videos at [https://sites.google.com/view/decisiondiffuser/](https://sites.google.com/view/decisiondiffuser/)). We discuss this baseline in Appendix I.2.
> > > > >
> > > > > >Sec. 3 “It is useful to solve RL from offline data, both without relying on TD-learning and without risking distribution-shift.” - what are the conditions/requirements for the latter to be possible (at least informally)? In order to not risk distribution shift strong conditions on training-data coverage seem required.
> > > > >
> > > > > We would like to clarify what we meant by *risking distribution-shift*. When doing offline RL with temporal difference (TD) learning naively, the learned policy $\pi(a|s)$ often predicts out-of-distribution (OOD) actions with erroneously high Q-value. This happens no matter the dataset size/coverage as there will always exist an OOD action with erroneously high Q-value, as noted in [4]. In theory, this can be prevented by constraining the learned policy $\pi(a|s)$ to be close to the behavior policy that generated the dataset. However, the resulting constrained TD optimization is unstable in practice, as noted in [7]. Hence, we run the risk of too much distribution shift between the learned policy and the behavior policy leading to instabilities when doing offline RL with TD learning. In contrast, Decision diffuser doesn't face the risk of distribution-shift between the learned state trajectory distribution and the empirical (dataset) state trajectory distribution as its trained on the dataset with maximum-likelihood estimation.
> > > > >
> > > > > Of course, any offline RL method, including Decision Diffuser, will face distribution shift between the learned state trajectory distribution and the true state trajectory distribution due to distribution shift between empirical (dataset) state trajectory distribution and true state trajectory distribution. This can only be reduced if the offline dataset has extensive coverage of the state-action space. This is more formally analyzed in [5].
> > > > >
> > > > > >P4, typo: “choices fof diffusion”
> > > > >
> > > > > We have fixed the typo in our updated draft.

---

> > > > > > ### Author Response · Authors · 2022-11-15
> > > > > > **Author's response (6/6)**
> > > > > >
> > > > > > **References**
> > > > > > 1. Classifier-Free Diffusion Guidance. Jonathan Ho, Tim Salimans. arXiv:2207.12598.
> > > > > > 2. GLIDE: Towards Photorealistic Image Generation and Editing with Text-Guided Diffusion Models. Alex Nichol, Prafulla Dhariwal, Aditya Ramesh, Pranav Shyam, Pamela Mishkin, Bob McGrew, Ilya Sutskever, Mark Chen. arXiv:2112.10741.
> > > > > > 3. Photorealistic Text-to-Image Diffusion Models with Deep Language Understanding. Chitwan Saharia, William Chan, Saurabh Saxena, Lala Li, Jay Whang, Emily Denton, Seyed Kamyar Seyed Ghasemipour, Burcu Karagol Ayan, S. Sara Mahdavi, Rapha Gontijo Lopes, Tim Salimans, Jonathan Ho, David J Fleet, Mohammad Norouzi. arXiv:2205.11487.
> > > > > > 4. Offline Reinforcement Learning: Tutorial, Review,
> > > > > > and Perspectives on Open Problems. Sergey Levine, Aviral Kumar, George Tucker, Justin Fu. arXiv:2005.01643.
> > > > > > 5. Conservative Q-Learning for Offline Reinforcement Learning. Aviral Kumar, Aurick Zhou, George Tucker, Sergey Levine. NeurIPS 2020.
> > > > > > 6. Off-Policy Deep Reinforcement Learning without Exploration. Scott Fujimoto, David Meger, Doina Precup. ICML 2019.
> > > > > > 7. RvS: What is Essential for Offline RL via Supervised Learning? Scott Emmons, Benjamin Eysenbach, Ilya Kostrikov, Sergey Levine. arXiv:2112.10751.
> > > > > > 8. Variational Diffusion Models. Diederik P. Kingma, Tim Salimans, Ben Poole, Jonathan Ho. NeurIPS 2021.
> > > > > > 9. Meta-Reinforcement Learning of Structured Exploration Strategies. Abhishek Gupta, Russell Mendonca, YuXuan Liu, Pieter Abbeel, Sergey Levine. arXiv:1802.07245.
> > > > > > 10. Soft Actor-Critic: Off-Policy Maximum Entropy Deep Reinforcement Learning with a Stochastic Actor. Tuomas Haarnoja, Aurick Zhou, Pieter Abbeel, Sergey Levine. ICML 2018.
> > > > > > 11. Online Decision Transformer. Qinqing Zheng, Amy Zhang, Aditya Grover. ICML 2022.
> > > > > > 12. Dichotomy of Control: Separating What You Can Control from What You Cannot. Mengjiao Yang, Dale Schuurmans, Pieter Abbeel, Ofir Nachum. arXiv:2210.13435.
> > > > > > 13. You Can't Count on Luck: Why Decision Transformers Fail in Stochastic Environments. Keiran Paster, Sheila McIlraith, Jimmy Ba. arXiv:2205.15967.
> > > > > > 14. Addressing Optimism Bias in Sequence Modeling for Reinforcement Learning. Adam R Villaflor, Zhe Huang, Swapnil Pande, John M Dolan, Jeff Schneider. ICML 2022.
> > > > > > 15. Decision Transformer: Reinforcement Learning via Sequence Modeling. Lili Chen^, Kevin Lu^, Aravind Rajeswaran, Kimin Lee, Aditya Grover, Michael Laskin, Pieter Abbeel, Aravind Srinivas+, Igor Mordatch+. NeurIPS 2021 (^ Equal contribution, + Equal advising).
> > > > > > 16. Offline Reinforcement Learning as One Big Sequence Modeling Problem. Michael Janner, Qiyang Li, Sergey Levine. NeurIPS 2021.

---

> ### Author Response · Authors · 2022-11-17
> **Looking forward to further discussions!**
>
> Dear Reviewer,
>
> Thank you for your time and effort in reviewing our work. We have provided detailed clarification and additional experiments to address the issues raised in your comments. If our response has addressed your concerns, we would be grateful if you could re-evaluate our work.
>
> If you have any additional questions or comments, we would be happy to have further discussions.
>
> Thanks,
>
> Authors

---

> > ### Comment · Reviewer_9svn · 2022-11-29
> > **Thank you for the extensive and very detailed response**
> >
> > Thank you for the very detailed response with many important and very interesting clarifications, and a number of insightful additional results and evaluations. It is fair to say that the issues raised by me have been addressed sufficiently, and I think the paper is stronger overall now and the potential impact is increased. Rather than continuing the detailed discussion in the comments below, I think the paper is now clearly above the acceptance threshold - thank you for the work and effort put into the rebuttal.

---

### Official Review · Reviewer_87NB · 2022-11-01

**Confidence:** 3
**Correctness:** 3
**Technical Novelty And Significance:** 2
**Empirical Novelty And Significance:** 2
**Recommendation:** 6

**Clarity, Quality, Novelty And Reproducibility:**

This paper is well motivated and easy to follow. Inspired from diffusion models in vision domain, this paper formulate the decision making process as a condition generation problem and which can naturally achieved by leveraging diffusion models. I like the way that this paper very clearly introducing the technical background and formulate the problem. From the experimental results, the proposed Decision Diffuser is promising on a couple of evaluating tasks.
My main concern is the limited technical novelty. The key contribution of this work is to formulate the decision making problems as a conditional generation problem. Based on this, this paper train a diffusion model on offline datasets. However, it is more like an application of diffusion models on decision making tasks. I don't see obvious novelty from either model design and/or training objectives.
The evaluations are weak. It only evaluate the DD variants on a few tasks. No comparisons of DD with the other state-of-the-art approaches are shown. Also, no ablation and discussion are shown. It is not convincing to me without such extensive study.
It did not submit the source code, so the reproducibility is hard to be validated.

**Strength And Weaknesses:**

Pros:
This paper is well motivated and easy to follow. Inspired from diffusion models in vision domain, this paper formulate the decision making process as a condition generation problem and which can naturally achieved by leveraging diffusion models. I like the way that this paper very clearly introducing the technical background and formulate the problem. From the experimental results, the proposed Decision Diffuser is promising on a couple of evaluating tasks.

Cos:
My main concern is the limited technical novelty. The key contribution of this work is to formulate the decision making problems as a conditional generation problem. Based on this, this paper train a diffusion model on offline datasets. However, it is more like an application of diffusion models on decision making tasks. I don't see obvious novelty from either model design and/or training objectives.
The evaluations are weak. It only evaluate the DD variants on a few tasks. No comparisons of DD with the other state-of-the-art approaches are shown. Also, no ablation and discussion are shown. It is not convincing to me without such extensive study.

**Summary Of The Paper:**

This paper presents Decision Diffuser, a conditional generative model for sequential decision making. It frames offline sequential decision making as conditional generative modeling by considering  two other variables: constraints and skills. Conditioning on a single constraint or skill during training leads to behaviors at test-time that can satisfy several constraints together or demonstrates a composition of skills. Experiments are conducted on a couple of different decision making tasks.

**Summary Of The Review:**

This paper is well motivated and easy to follow. Inspired from diffusion models in vision domain, this paper formulate the decision making process as a condition generation problem and which can naturally achieved by leveraging diffusion models. I like the way that this paper very clearly introducing the technical background and formulate the problem. From the experimental results, the proposed Decision Diffuser is promising on a couple of evaluating tasks.
My main concern is the limited technical novelty. The key contribution of this work is to formulate the decision making problems as a conditional generation problem. Based on this, this paper train a diffusion model on offline datasets. However, it is more like an application of diffusion models on decision making tasks. I don't see obvious novelty from either model design and/or training objectives.
The evaluations are weak. It only evaluate the DD variants on a few tasks. No comparisons of DD with the other state-of-the-art approaches are shown. Also, no ablation and discussion are shown. It is not convincing to me without such extensive study.
In addition, it did not submit the source code, so the reproducibility is hard to be validated.

---

> ### Author Response · Authors · 2022-11-15
> **Author's Response (1/2)**
>
> We thank reviewer 87NB for evaluating our work. We now answer the following concerns raised in the review.
>
> >My main concern is the limited technical novelty
>
> Our submission leverages recent developments in diffusion generative modeling to improve offline decision-making and makes non-trivial design choices that aren't immediately obvious:
>
> - **Classifier-free guidance and low-temperature sampling instead of dynamic programming** Rather than learn a Q function from the offline dataset to guide a policy towards optimal actions, we avoid TD learning altogether (and the instabilities [1] that come with it) and use classifier-free guidance to implicitly guide our policy towards optimal actions. Furthermore, low temperature sampling makes it easier to sample the mode of the conditional state trajectory distribution (conditioned on optimal return) and thus obtain high rewarding state trajectories. We empirically justify these design choices through our experiments in Section 4.1 (Table 1) and Appendix C.
>
> - **Inverse dynamics for extracting actions instead of diffusing over actions** As mentioned in Section 3.1, sequences over actions, which are often represented as joint torques, tend to be more high-frequency and less smooth, making them much harder to predict and model with diffusion (as noted in [2]). Hence, we choose to only diffuse over states and extract actions with learned inverse dynamics model. We empirically justify this design choice through our ablation study in Section 4.1 (Table 2). Upon suggestion of reviwer 9svn, we further empirically analyze when to use inverse dynamics for actions vs when to diffuse over actions in Appendix F.
>
> - **Satisfying constraints and composing skills with diffusion** A diffusion model learns the score function of the underlying (conditional) data distribution. Hence, Decision diffuser can sample from composed conditional trajectory distribution $q(x_0(\tau)| (y^i(\tau))_{i=1}^n)$ by combining predictions of noise model $\epsilon_\theta$ conditioned on different variables $y^i(\tau)$:
>
>      $\hat{\epsilon} = \epsilon_\theta(x_t(\tau), \emptyset, t) + s\sum_{i=1}^n (\epsilon_\theta(x_t(\tau), y^i(\tau), t) - \epsilon_\theta(x_t(\tau), \emptyset, t))$
>
>      We use this to combine constraints (Section 4.2) or compose skills (Section 4.3) flexibly during test time. We explain this composition in detail in Appendix D. Additionally, we show that Decision Diffuser can support "NOT" compoistions, in addition to "AND" compositions, in Appendix J.
>
> In summary, the novelty of our submission lies in non-trivial design choices we made while bringing diffusion modeling to offline decision making and in leveraging the score-based (or energy based) interpretation of diffusion models to combine constraints and compose skills flexibly, thereby generating new behaviors during test time.
>
> >It only evaluate the DD variants on a few tasks. No comparisons of DD with the other state-of-the-art approaches are shown.
>
> **Lack of comparison with other state-of-the-art approaches** We respectfully disagree with this characterization and the above statement isn't true. In our original submission (Section 4.1, Table 4), for offline RL, we compared Decision Diffuser to high-quality SOTA approaches like Conservative Q-Learning (CQL) [3], Implicit Q-Learning (IQL) [4], Decision Transformer (DT) [5], Trajectory Transformer (TT) [6], MoREL [7] and Diffuser [8]. For constraint satisfaction, we compared Decision Diffuser to Diffuser [8], BCQ [9] and CQL [3].
>
> Furthermore, our view is corroborated by reviewer 9svn who noted that our "empirical results show benefits of proposed method compared to high-quality SOTA methods on standard benchmarks" and reviewer WRRm who noted our "performance on par with other complex reinforcement learning".
>
> **Evaluation on few tasks** In our original submission, we evaluated Decision Diffuser on 11 D4RL tasks, 4 kuka block stacking tasks and on unitree go environment. In our updated draft, we have added experiments on Block push tasks where we compare performance of Decision Diffuser with those of Diffuser and CQL while varying the stochasticity in environment dynamics (Appendix G).

---

> > ### Author Response · Authors · 2022-11-15
> > **Author's Response (2/2)**
> >
> > >Also, no ablation and discussion are shown.
> >
> > We again respectfully disagree with this characterization. While reviewers are welcome to criticize us over the lack of particular ablation studies, it is false to say that we had no ablation.
> >
> > In our original submission, we showed the importance of inverse dynamics by comparing Decision Diffuser to CondDiffuser baseline that diffuses over both state and action (Section 4.1, Table 2). Furthermore, we showed how low-temperature sampling was important for the performance of Decision Diffuser as it leads to less variance in generated trajectories and better captures the mode of the underlying conditional trajectory distribution (Appendix C).
> >
> > During the rebuttal phase, upon suggestions of reviewers 9svn and pyhf, we added following ablation studies:
> > 1. **Runtime characteristics of Decision Diffuser**: We discuss the runtime characteristics of Decision Diffuser and evaluate its performance while varying the number of reverse diffusion steps in Appendix E.
> > 2. **When to use inverse dynamics for extracting actions**: We empirically analyze when to use inverse dynamics for extracting actions vs when to diffuser over actions in Appendix F.
> > 3. **Robustness under stochastic dynamics**: We evaluate the robustness of Decision Diffuser and other baselines (Diffuser, CQL) as we vary stochasticity in environment dynamics in Appendix G.
> > 4. **Simple baseline for skill composition**: In Appendix I.2, we compare skill composition via score function addition (Equation 9) to a simple baseline where we condition noise model $\epsilon_\theta$ on sum of conditioning variables (represented as one-hot vectors) corresponding to skills that are being composed.
> >
> > **References**
> > 1. Offline Reinforcement Learning: Tutorial, Review,
> > and Perspectives on Open Problems. Sergey Levine, Aviral Kumar, George Tucker, Justin Fu. arXiv:2005.01643.
> > 2. Variational Diffusion Models. Diederik P. Kingma, Tim Salimans, Ben Poole, Jonathan Ho. NeurIPS 2021.
> > 3. Conservative Q-Learning for Offline Reinforcement Learning. Aviral Kumar, Aurick Zhou, George Tucker, Sergey Levine. NeurIPS 2020.
> > 4. Offline Reinforcement Learning with Implicit Q-Learning. Ilya Kostrikov, Ashvin Nair, Sergey Levine. ICLR 2022.
> > 5. Decision Transformer: Reinforcement Learning via Sequence Modeling. Lili Chen^, Kevin Lu^, Aravind Rajeswaran, Kimin Lee, Aditya Grover, Michael Laskin, Pieter Abbeel, Aravind Srinivas+, Igor Mordatch+. NeurIPS 2021 (^ Equal contribution, + Equal advising).
> > 6. Offline Reinforcement Learning as One Big Sequence Modeling Problem. Michael Janner, Qiyang Li, Sergey Levine. NeurIPS 2021.
> > 7. MOReL: Model-Based Offline Reinforcement Learning. Rahul Kidambi, Aravind Rajeswaran, Praneeth Netrapalli, Thorsten Joachims. NeurIPS 2020.
> > 8. Planning with Diffusion for Flexible Behavior Synthesis. Michael Janner^, Yilun Du^, Joshua Tenenbaum, and Sergey Levine. ICML 2022 (^ Equal contribution).
> > 9. Off-Policy Deep Reinforcement Learning without Exploration. Scott Fujimoto, David Meger, Doina Precup. ICML 2019.

---

> > > ### Comment · Reviewer_87NB · 2022-12-01
> > > **Response to authors**
> > >
> > > Thanks for the authors' response. My concerns and questions are pretty much addressed in the rebuttal.
> > > I do appreciate the authors' detailed clarification and discussion. Considering these, I would like to raise my score to 6.

---

> ### Author Response · Authors · 2022-11-17
> **Looking forward to further discussions!**
>
> Dear Reviewer,
>
> Thank you for your time and effort in reviewing our work. We have provided detailed clarification and additional experiments to address the issues raised in your comments. If our response has addressed your concerns, we would be grateful if you could re-evaluate our work.
>
> If you have any additional questions or comments, we would be happy to have further discussions.
>
> Thanks,
>
> Authors

---

### Author Response · Authors · 2022-11-15
**Common Response**

We thank the reviewers for their thoughtful suggestions and comments. We have run some new experiments and added visualizations to help address the concerns brought up by the reviewers, which we summarize below. These changes can be found in the updated revision, highlighted in purple color, and described below.

1. **Quantifying skill composition for unitree environment**: We learn and make use of a gait classifier to quantify how well different gaits are being composed. We expand Section 4.3 to include the gait classification results and further detail this in Appendix I and Appendix I.1.

2. **Simple baseline for skill composition**: In Appendix I.2, we compare skill composition via score function addition (Equation 9) to a simple baseline where we condition noise model $\epsilon_\theta$ on sum of conditioning variables (represented as one-hot vectors) corresponding to skills that are being composed.

3. **Discussion on assumptions for composition of conditioning variables** We add the assumptions required for composition of conditioning variables in Section 3.3 and detail this in Appendix D where we show how these assumptions lead to derivation of Equation 9 in the paper.

4. **Support for NOT compositions** We show how Decision Diffuser supports NOT composition in Appendix J.

5. **Runtime characteristics of Decision Diffuser**: We discuss the runtime characteristics of Decision Diffuser and evaluate its performance while varying the number of reverse diffusion steps in Appendix E.

6. **When to use inverse dynamics for extracting actions**: We empirically analyze when to use inverse dynamics for extracting actions vs when to diffuser over actions in Appendix F.

7. **Robustness under stochastic dynamics**: We evaluate the robustness of Decision Diffuser and other baselines (Diffuser, CQL) as we vary stochasticity in environment dynamics in Appendix G.

8. **Comparing Q-function guidance and Classifier-free guidance** We compare and discuss Q-function guided diffusion and Classifier-free guided diffusion in Appendix K.

9. **Limitations of Decision Diffuser** We summarize limitations of Decision Diffuser in Appendix L.

We now individually address the concerns of reviewers. Please see responses below each review.

---

### Decision · Program_Chairs · 2023-01-20

**Decision:**

Accept: notable-top-5%

**Justification For Why Not Higher Score:**

N/A

**Justification For Why Not Lower Score:**

I think this paper will be of broad interest to the ICLR community (both on the RL and generative models side) and as such would be well suited to an oral presentation. The paper is clearly written, with many interesting ablations incorporated during the rebuttal phase, which means everyone will likely find something of interest and something to learn from the talk.

An oral presentation would also give exposure to the high-level question posed by the paper's title: is conditional generative modeling enough for RL? Regardless of the answer, these models are clearly powerful and the RL community should pay attention. An oral is the best way to do this, and hopefully leads to some of these advances being incorporated in more classical online approaches based on temporal difference learning.

**Metareview: Summary, Strengths And Weaknesses:**

This paper introduces the Decision Diffuser (DD) model, a novel application of diffusion models to offline reinforcement learning. Key to their approach is the use of classifier-free guidance for conditioning on returns, skills or constraints, which allows for dynamic recombination of behaviors at test time and the incorporation of an inverse dynamics model for generating actions from generated state sequences.

Reviewers [9svn, pyhf, WRRm, AC] agree that the method is shown to be effective across a broad range of challenging environments, and baselines spanning classical offline RL methods, and other recent advances based on conditional generative modeling (Decision Transformer, Diffuser). Furthermore, the paper is clearly written and the revised manuscript has incorporated some notable improvements (clarifications, and many ablations) which address most if not all of the reviewers’ concern. Well done! In particular, I commend the authors’ for their clear and concise explanation of diffusion models and classifier-free guidance which will be most useful to the RL community.

As mentioned by [pyhf], this is an exciting and timely contribution which will be of great interest to ICLR: both to the general RL community, but also the generative models crowd as another successful application of diffusion models. For the RL crowd, not online does this paper offer a convincing alternative to offline RL, it also adds significant weight to the general question posed by the papers’ title. Perhaps conditional generative modeling is enough? Either way, I’m excited to see where this work leads us. Congratulations to the authors!


**Note From Pc:**

if the above contains the word "oral" or "spotlight" please see: "oral" presentation means -> notable-top-5% and "spotlight" means -> notable-top-25%. As stated in our emails, we are disassociating presentation type from AC recommendations